# In vitro reconstitution reveals major differences between human and bacterial cytochrome c synthases

Molly C Sutherland[1,2†]*, Deanna L Mendez[1†], Shalon E Babbitt[1†‡],
Dustin E Tillman[1§], Olga Melnikov[1], Nathan L Tran[1], Noah T Prizant[1],
Andrea L Collier[1], Robert G Kranz[1]*

[1]Department of Biology, Washington University in St. Louis, St. Louis, United States;
[2]Department of Biological Sciences, University of Delaware, Newark, United States

**Abstract** Cytochromes c are ubiquitous heme proteins in mitochondria and bacteria, all possessing a CXXCH (CysXxxXxxCysHis) motif with covalently attached heme. We describe the first in vitro reconstitution of cytochrome c biogenesis using purified mitochondrial (HCCS) and bacterial (CcsBA) cytochrome c synthases. We employ apocytochrome c and peptide analogs containing CXXCH as substrates, examining recognition determinants, thioether attachment, and subsequent release and folding of cytochrome c. Peptide analogs reveal very different recognition requirements between HCCS and CcsBA. For HCCS, a minimal 16-mer peptide is required, comprised of CXXCH and adjacent alpha helix 1, yet neither thiol is critical for recognition. For bacterial CcsBA, both thiols and histidine are required, but not alpha helix 1. Heme attached peptide analogs are not released from the HCCS active site; thus, folding is important in the release mechanism. Peptide analogs behave as inhibitors of cytochrome c biogenesis, paving the way for targeted control.

*For correspondence:
msuther@udel.edu (MCS);
Kranz@wustl.edu (RGK)

†These authors contributed equally to this work

Present address: ‡Pfizer, Chesterfield, United States;
§Department of Molecular and Cellular Biology, Harvard University, Cambridge, United States

## Introduction

The structure of cytochrome c (cyt c), as well as its key function in electron transport for aerobic respiration, have been known for over half a century (*Dickerson et al., 1971*; *Ernster and Schatz, 1981*). Scores of newly discovered and extraordinary electron transport chains with unique cyt c proteins in bacteria are now known, such as extracellular multiheme nanowires comprised of many c-type hemes (e.g. *Deane, 2019*; *Wang et al., 2019*). In addition to its role in respiration, cyt c is known to play other important functions, such as activation of programmed cell death in eukaryotes (apoptosis) (*Ow et al., 2008*; *Tait and Green, 2010*). Regardless of its function, each c-type heme contains two thioether attachments to a conserved CysXxxXxxCysHis (CXXCH) motif, where the histidine acts as an axial ligand to the heme iron in the native cyt c (*Figure 1—figure supplement 1a, b*; *Dickerson et al., 1971*). It is generally agreed that the covalently attached heme makes these energy conversion proteins particularly stable (e.g. *Allen et al., 2005*). In fact, recent engineering of novel and stable heme-based catalysts has used c-heme polypeptides produced in vivo (*Kan et al., 2017*; *Kan et al., 2016*; *Watkins et al., 2017*).

   To form c-heme, heme is attached stereochemically to each CXXCH motif and it appears that in the case of cyt c, folding into its native structure occurs after attachment (*Kranz et al., 2009*). Cyt c biogenesis requires accessory proteins that are needed to attach the heme group and complete maturation. Three pathways have been discovered and characterized genetically, called Systems I, II, III (*Figure 1—figure supplement 1b,c*) (reviewed in *Kranz et al., 2009*; *Ferguson et al., 2008*; *Kranz et al., 1998*; *Bowman and Bren, 2008*; *Simon and Hederstedt, 2011*; *Verissimo and Daldal, 2014*; *Gabilly and Hamel, 2017*). Systems I and II have evolved in bacteria, while System III is in

**eLife digest** From tiny bacteria to the tallest trees, most life on Earth carries a protein called cytochrome c, which helps to create the energy that powers up cells. Cytochrome c does so thanks to its heme, a molecule that enables the chemical reactions required for the energy-creating process.

Despite both relying on cytochrome c, animals and bacteria differ in the enzyme they use to attach the heme to the cytochrome. Spotting variations in how this 'cytochrome c synthase' works would help to find compounds that deactivate the enzyme in bacteria, but not in humans. However, studying cytochrome c synthase in living cells is challenging.

To bypass this issue, Sutherland, Mendez, Babbitt et al. successfully reconstituted cytochrome c synthases from humans and bacteria in test tubes. This allowed them to examine in detail which structures the enzymes recognize to spot where to attach the heme onto their target. The experiments revealed that human and bacterial synthases actually rely on different parts of the cytochrome c to orient themselves. Different short compounds could also block either the human or bacterial enzyme.

Variations between human and bacterial cytochrome c synthase could lead to new antibiotics which deactivate the cytochrome and kill bacteria while sparing patients. The next step is to identify molecules that specifically interfere with cytochrome c synthase in bacteria, and could be tested in clinical trials.

most mitochondria. Each system possesses a cyt c synthase (*Figure 1—figure supplement 1c*, orange), which attaches the two vinyl groups of heme to cysteines of CXXCH. However, the cyt c biogenesis process, starting with CXXCH recognition, to heme attachment, to release and final folding, remains largely unknown. While in vivo studies have suggested some requirements (*Babbitt et al., 2017*; *Babbitt et al., 2016*; *Corvest et al., 2010*; *San Francisco et al., 2013*), such cyt c genetic studies do not examine problems of instability, recognition, release, or folding of the cyt c variants. Direct testing of substrates without these limitations awaited the development of in vitro reconstitution. The mitochondrial System III is composed of a cyt c synthase called HCCS (holo-cyt c synthase) in the intermembrane space (*Figure 1a* and *Figure 1—figure supplement 1c*, *Pollock et al., 1998*; *Dumont et al., 1987*; *Babbitt et al., 2015*). Bacterial systems are unrelated to HCCS and more complicated, heme attachment occurs 'outside' the cells; thus, these pathways export the heme and attach it to secreted, unfolded cyt c. System II is composed of a large integral membrane protein complex called CcsBA (*Beckett et al., 2000*; *Dreyfuss et al., 2003*; *Xie and Merchant, 1996*) (sometimes called ResBC [*Ahuja et al., 2009*; *Le Brun et al., 2000*]), which is proposed to both export heme and then attach it to cyt c CXXCH motifs (*Feissner et al., 2006*; *Frawley and Kranz, 2009*). Specific factors for thiol reduction of the CXXCH motifs have also been proposed (*Bonnard et al., 2010*; *Kranz et al., 2009*).

Large gaps in the cyt c biogenesis field remain such as CXXCH recognition requirements by each cyt c synthase and whether other general factors in the cell are needed for recognition, heme attachment, and folding. While specific proteins have been identified and functions hypothesized for each system (reviewed in *Babbitt et al., 2015*; *Ferguson et al., 2008*; *Gabilly and Hamel, 2017*; *Kranz et al., 2009*; *Verissimo and Daldal, 2014*), there has been no in vitro reconstitution studies with purified cyt c synthases, which will be needed to address these gaps. Only recently was our group able to purify the cyt c synthases, after recombinant expression in *Escherichia coli* (*Frawley and Kranz, 2009*; *Merchant, 2009*; *Richard-Fogal et al., 2009*; *San Francisco et al., 2013*; *Sutherland et al., 2018b*). Here we develop and characterize the first in vitro reconstitutions of cyt c synthases, using purified human HCCS and the bacterial CcsBA. No protein factors other than the cyt c synthases are needed in vitro for attachment and folding into a native cyt c structure. In vitro reactions with a variety of peptides containing CXXCH show that the CXXCH substrates for each cyt c synthase are quite different and that post-attachment folding of cyt c is important in release from the synthase active sites. Key differences between HCCS and CcsBA include thiol (cysteine) requirements and the alpha helix sequence adjacent to CXXCH. Peptide analogs behave as inhibitors. Because bacteria and humans (mitochondria) use very different cyt c synthases, shown

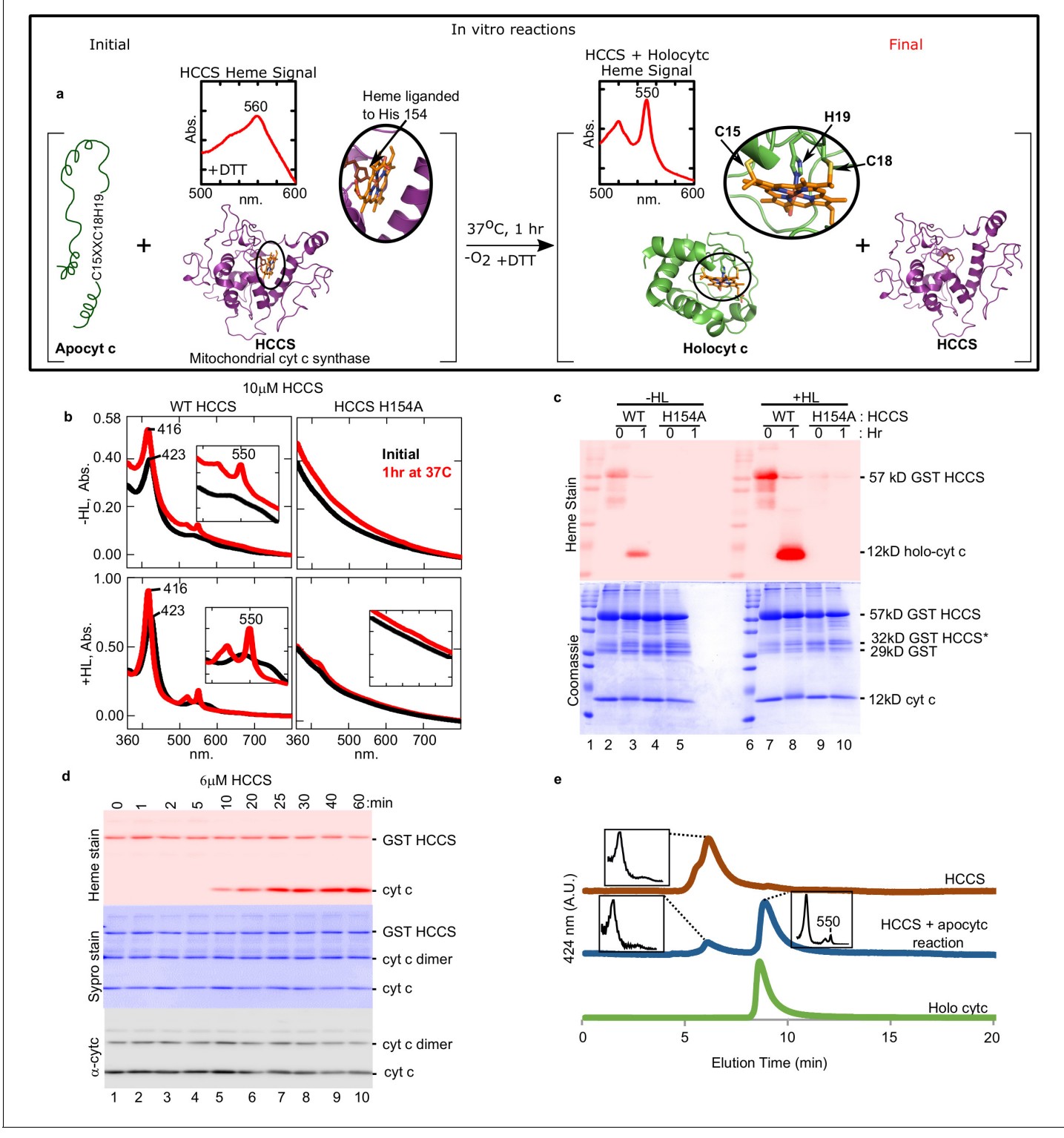

**Figure 1.** Cyt c is biosynthesized in vitro by mitochondrial HCCS. (a) Schematic of the in vitro heme attachment reaction of HCCS with apocytochrome c (apocyt c). Mitochondrial cyt c synthase, HCCS, positions heme (orange) and attaches it to apocyt c. Cyt c is released and folds into its native structure. Insets show the UV–vis spectra of heme. (b) UV–vis spectra of heme signal from the anaerobic reaction of WT and H154A HCCS (±heme loading [HL]) with apocyt c as outlined in a black line – initial, red line – 1 hr post-addition of DTT. Inset shows magnification of the $\beta/\alpha$ region. (c) In vitro biosynthesis of cyt c was monitored by heme stain. WT HCCS biosynthesized 12 kDa cyt c product (lanes 3 and 8). HCCS H154A, a mutant defective for heme binding, did not (lanes 5 and 10). Total protein for in vitro reaction shown by Coomassie. For (b) and (c), representative data is

*Figure 1 continued*

shown from three biological replications (independent purifications of HCCS). (**d**) Time course of HCCS in vitro activity. A single trial showed heme-stained cyt c product is first observed after 10 min (red, lane 5). Sypro stain shows total protein levels, α-cyt c shows total cyt c in reaction. The apocyt c dimer observed upon SDS–PAGE is due to aggregation and does not impact the results or conclusions. (**e**) HPLC profiles of the indicated reaction products representative of two trials.

The online version of this article includes the following figure supplement(s) for figure 1:

**Figure supplement 1.** Cyt c attachment to heme and cyt c biogenesis pathways (Systems I, II, and III).

**Figure supplement 2.** Titration of heme loading (HL) in GST-HCCS.

**Figure supplement 3.** HCCS in vitro biosynthesized cyt c has heme attached and is properly folded.

**Figure supplement 4.** In-vitro biosynthesis of cyt c by HCCS is temperature dependent.

**Figure supplement 5.** HCCS function is dependent on the presence of DTT (aerobic conditions).

**Figure supplement 6.** Aerobic in vitro reaction with HCCS + apocyt c.

**Figure supplement 7.** Model for HCCS function proposed previously based on in vivo results (*Babbitt et al., 2015*; *San Francisco et al., 2013*).

here to recognize distinct features of the CXXCH substrate, specific inhibitors could constitute targeted antimicrobials, facilitating chemical control of cyt c levels in selected organisms.

## Results

### In vitro reconstitution of HCCS using apocyt c as substrate

Using purified human HCCS, we reconstituted cyt c synthase activity with equine apocyt c as substrate, initially assaying formation of a peak at 550 nm, diagnostic of cyt c's typical UV–vis spectra (*Figure 1a*). Recombinant human HCCS (GST-tagged) is functional in vivo, attaching heme to co-expressed apocyt c (*San Francisco et al., 2013*) in *E. coli*. We have previously shown that HCCS co-purifies with heme, which is liganded to His154 (*San Francisco et al., 2013*). UV–vis spectra of purified HCCS shows a 423 nm and broad 560 nm absorption, typical of heme proteins, while HCCS H154A variant does not bind heme (*Figure 1b*, −HL). We developed a 'heme-loading (HL)' protocol to increase the levels of heme bound in HCCS (+HL, ~30% occupied) above the co-purified levels of endogenous heme (−HL, ~10% occupied). HL was also advantageous since the HL protocol removes excess heme, thus minimizing spectral interference from free heme in reactions. HL was shown to depend on the natural His154 ligand (*Figure 1b*, +HL black line), and loading was saturated at 2–5 µM heme (*Figure 1—figure supplement 2*). Initial reconstitutions were performed with wild type (wt) HCCS (±HL) and the HCCS His154Ala variant that does not bind heme (*Figure 1b*). Upon incubation for 1 hr in the presence of apocyt c and dithiothreitol (DTT), a sharp 550 nm peak emerged, indicative of a c-type cytochrome (*Figure 1b*, red with wt). This occurred with wt HCCS containing endogenous heme (−HL) and in vitro loaded heme (+HL), while HCCS H154A did not produce the 550 nm peak. A second method to determine if heme has been covalently attached to the apocyt c is to separate reactions with denaturing sodium dodecyl sulphate–polyacrylamide gel electrophoresis (SDS–PAGE) followed by heme staining, whereby covalently attached heme electrophoreses with the polypeptide (*Figure 1c*). Reactions with wt HCCS (−HL and +HL) and apocyt c confirmed that heme is covalently attached to cyt c (12 kDa) in the 1 hr reaction (*Figure 1c*, lanes 3, 8). As expected, no cyt c was formed with the HCCS H154A variant (*Figure 1c*, lanes 5, 10). Pyridine hemochrome spectra is often used to determine if two, one, or no covalent bonds to heme are present, with two thioether bonds showing a 550 nm peak (c-heme) and 560 nm for none (b-heme). The in vitro synthesized product has two thioether bonds, indicated by a 550 nm peak in pyridine hemochrome spectra (*Figure 1—figure supplement 3a*).

In vitro reconstitutions were studied for optimal conditions and requirements. Synthesis is optimal at 37°C (*Figure 1—figure supplement 4*), required DTT (*Figure 1—figure supplement 5*), with the cyt c product observed in 10 min (e.g. *Figure 1d*, lane 5). While cyt c is formed in both aerobic (*Figure 1—figure supplement 6*) and anaerobic conditions (*Figure 1b,c*), we decided to use anaerobic conditions for all studies since peptide substrates (below) under aerobic conditions required varying DTT concentrations, likely due to distinct thiol reducing requirements of individual peptides in air.

To further characterize HCCS, substrates and products, we employed analytical HPLC size exclusion chromatography (SEC), whereby UV–vis spectra of each separated species was recorded

(*Figure 1e*). HCCS (brown profile) elutes earlier than cyt c (green profile), and because these are 424 nm (heme) profiles, it is observed in the reaction (blue profile) that heme in HCCS decreases while cyt c product increases. These results also demonstrate that the cyt c product is released from the HCCS active site since it elutes at the same time as purified cyt c (holocyt c). We conclude that we have recapitulated in vitro the four-step process proposed previously (*Figure 1—figure supplement 7*) for HCCS-mediated cyt c biogenesis: heme binding (step 1), apocyt c binding (step 2), thioether formation (step 3), and release (step 4) (*Babbitt et al., 2015*; *San Francisco et al., 2013*). Next, we further characterize the released cyt c product to establish whether proper folding to the native state resulted from in vitro biogenesis.

We developed a HCCS-tethered (to glutathione beads) release assay to isolate HCCS reaction product(s), confirm that cyt c is released, and obtain high yields for product characterization (*Figure 2a*). Spectra of the released product (*Figure 2b*) is identical to holocyt c. SDS–PAGE of stages in the bead release protocol (*Figure 2c*) showed a released product of 12 kD that heme stained and reacted with cyt c antisera (*Figure 2c*, lane 2). We determined spectrally that the released cyt c has folded properly, forming the Met81 ligand as well as His19 (*Figure 1—figure supplement 3b*). Redox titrations (*Figure 2d*) showed that the redox potential of the cyt c in vitro product is the same as cyt c produced in vivo, +253 mV (*Figure 2d*). Analyses of supernatants (released), washes, and bead-retained material allowed for an estimate that at least 62% of cyt c is released

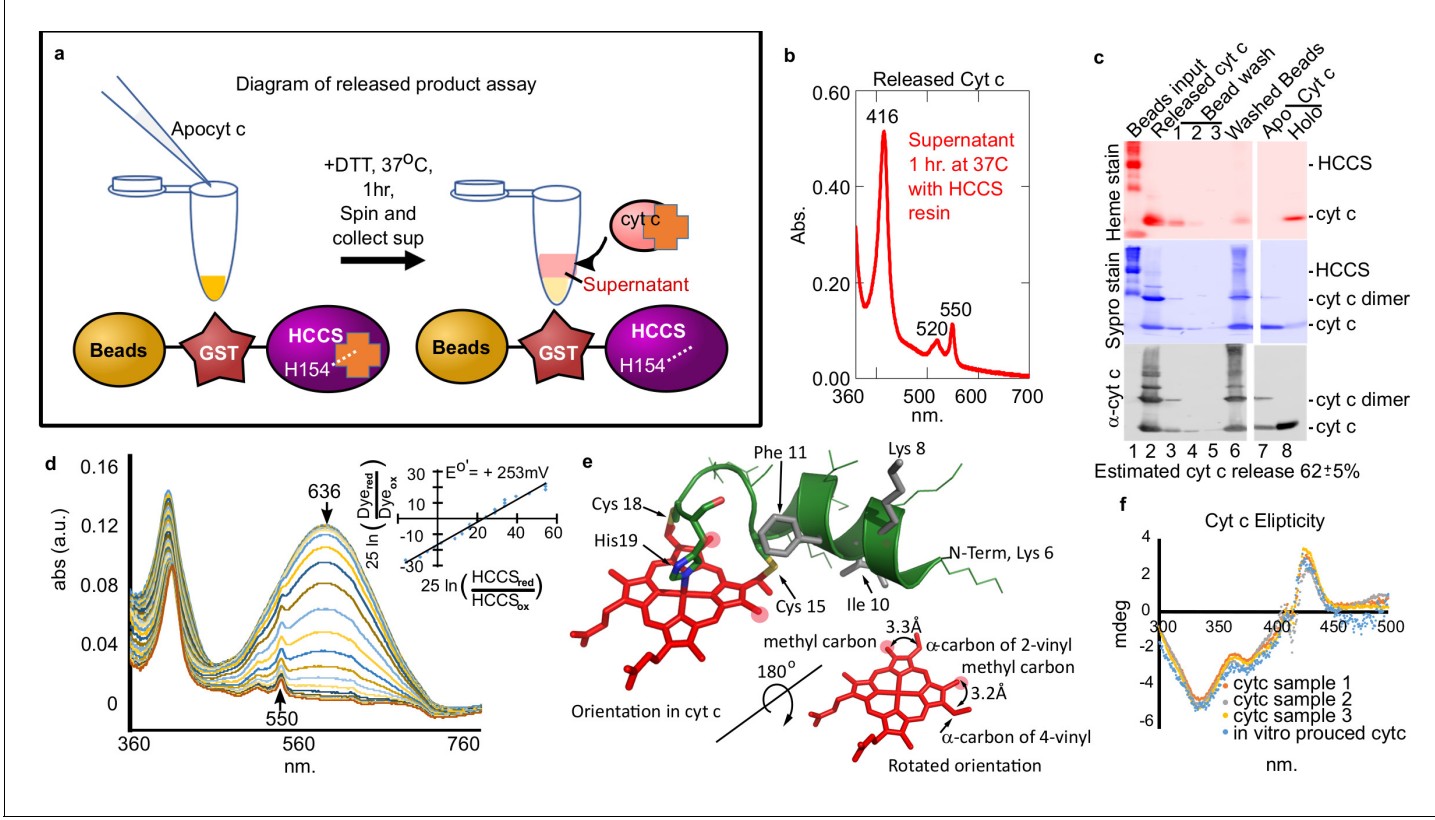

**Figure 2.** Cyt c biosynthesized in vitro is released by mitochondrial HCCS. (**a**) Schematic of HCCS released product assay. In vitro reaction is carried out with bead tethered GST-HCCS. Centrifugation separates the beads (GST-HCCS) and supernatant (e.g. released products). (**b**) UV–vis spectra of supernatant from the released product assay shows characteristic 550 nm cyt c peak, indicating cyt c is matured and released from GST-HCCS beads. (**c**) SDS–PAGE analysis of released product assay fractions. Lane two shows released cyt c as compared to purified holocyt c (lane 8). (**b**) and (**c**) are representative of three biological replicates. The standard deviation is provided. (**d**) The redox potential of the released cyt c was determined by a modified Massey method (*Efimov et al., 2007*) and determined to be +253 mV, similar to the published value for cyt c. This is data from one of three biological replicates. (**e**) Schematic of heme attached to cyt c from PDB: 3ZCF with heme rotated 180° (from *Babbitt et al., 2015*). (**f**) Circular dichroism (CD) spectra of in vivo (orange, gray, yellow) vs in vitro (blue) biosynthesized cyt c. In vivo cyts c represent three independent preparations.

© 2015, Elsevier permissions. Panel e is reproduced with permission from Figure 1, *Babbitt et al., 2015*, with permission from Elsevier. It is not covered by the CC-BY 4.0 licence and further reproduction of this figure would need permission from the copyright holder.

from HCCS (*Figure 2c*). Since heme in all cyt c's is attached stereochemically (*Figure 2e*), we performed circular dichroism (CD) spectra to compare the released (in vitro) product to cyt c made in vivo (*Figure 2f*). CD absorption of heme (~420 nm) is reduced in globins when heme binds in multiple orientations compared to a single orientation (*Aojula et al., 1986*; *Nagai et al., 2014*). Cyt c synthesized in vitro by HCCS shows an identical CD spectral profile as in vivo synthesized (*Figure 2f*). We conclude that in vitro reconstitution with purified HCCS results in stereochemical heme attachment, release, and proper folding of cyt c.

## Peptide analogs of apocyt c are recognized by HCCS and heme is covalently attached

In vitro reconstitution of the cyt c synthases provides an opportunity to investigate chemically synthesized apocyt c peptides and analogs as substrates. For example, there are in vivo genetic results suggesting that alpha helix 1, adjacent to the CXXCH motif (*Figure 3a*), of native cyt c is necessary for maturation by HCCS (*San Francisco et al., 2013*; *Zhang et al., 2014*; *Kleingardner and Bren, 2011*). In fact, the bacterial cyt c has a natural deletion of Met13 in alpha helix 1, recently shown in vivo to be the basis for the inability of HCCS to mature bacterial cyt c (*Babbitt et al., 2016*; *Verissimo et al., 2012*). We wanted to determine if cyt c peptides are recognized in vitro and if so the minimal sequence for recognition and heme attachment. Initially, we examined three peptides, an 11mer, 16mer, and 20mer with the 11mer lacking the sequence of alpha helix 1 (*Figure 3a*).

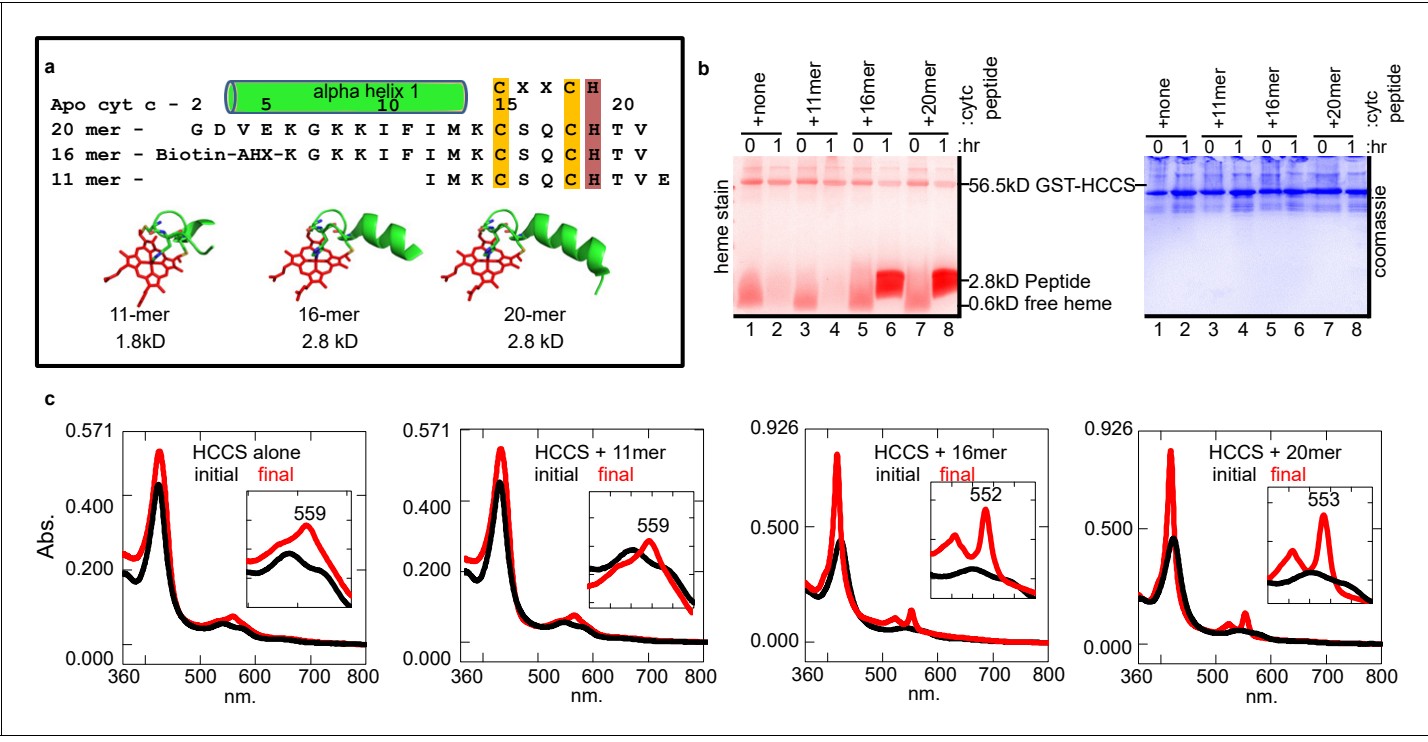

**Figure 3.** HCCS requires alpha helix 1 of cyt c for heme attachment to peptides containing CXXCH. (a) Sequence of three CXXCH containing peptides with alpha helix 1 and CXXCH designated. Three-dimensional structures of peptides with heme were generated from the cyt c 3D crystal structure PDB: 3ZCF, alpha helical structure is predicted, but not experimentally confirmed. In vitro reaction (as in *Figure 1a*) of HCCS and the peptides in 3 (a) was performed and analyzed by (b) SDS–PAGE followed by heme stain and (c) UV–vis spectra to assess heme signal. Black – initial, red – 1 hr post-addition of DTT. Inset shows magnification of the $\beta/\alpha$ region. Data is representative of three biological replicates.

The online version of this article includes the following source data and figure supplement(s) for figure 3:

**Figure supplement 1.** Attachment of heme to peptides by HCCS.
**Figure supplement 1—source data 1.** ImageJ Pixel analysis of heme stained bands.
**Figure supplement 2.** HCCS anaerobic in vitro attachment of heme to CXXCH variant peptides.
**Figure supplement 3.** In vitro HCCS attachment of heme to C15S 20mer and analysis of release.
**Figure supplement 3—source data 1.** Summary of percent released substrate.

Heme stains of tricine SDS–PAGE were used to detect whether heme was covalently attached to peptides (*Figure 3b*). After 1 hr, reactions showed that the 16mer (*Figure 3b*, lane 6) and 20mer (*Figure 3b*, lane 8) possessed an intense heme-stained peptide of 2.8 kDa, whereas the 11mer did not (*Figure 3b*, lane 4). Spectral analyses showed that the 11mer reaction looked like HCCS alone (no peptide added), whereas the reactions with the 16mer and 20mer showed a 552–553 nm peak (*Figure 3c*). We have previously shown that some recombinant HCCS is co-purified with cyt c remaining bound (and heme attached) (*San Francisco et al., 2013*). UV/vis absorption of these HCCS/cyt c complexes exhibits a peak in the reduced state of 553–555 nm (*Figure 1—figure supplement 7*), whereas a purified heme attached peptide shows a 550 nm peak (*Figure 1—figure supplement 7b*). Spectral results of HCCS reactions with 16mer and 20mer peptides (i.e. 552–553 nm peaks, see *Figure 3c*) suggest heme is covalently attached to the peptides, but that they remain in complex with HCCS, unlike full-length cyt c produced in vitro. To further test CXXCH peptide recognition, we tested a 56mer (with alpha helix 1 and 2 of cyt c) and a 9mer (*Figure 3—figure supplement 1*). While the 56mer was recognized and heme attached, the 9mer was not, consistent with the in vivo results that alpha helix 1 is required for heme attachment (*Babbitt et al., 2016*). Because

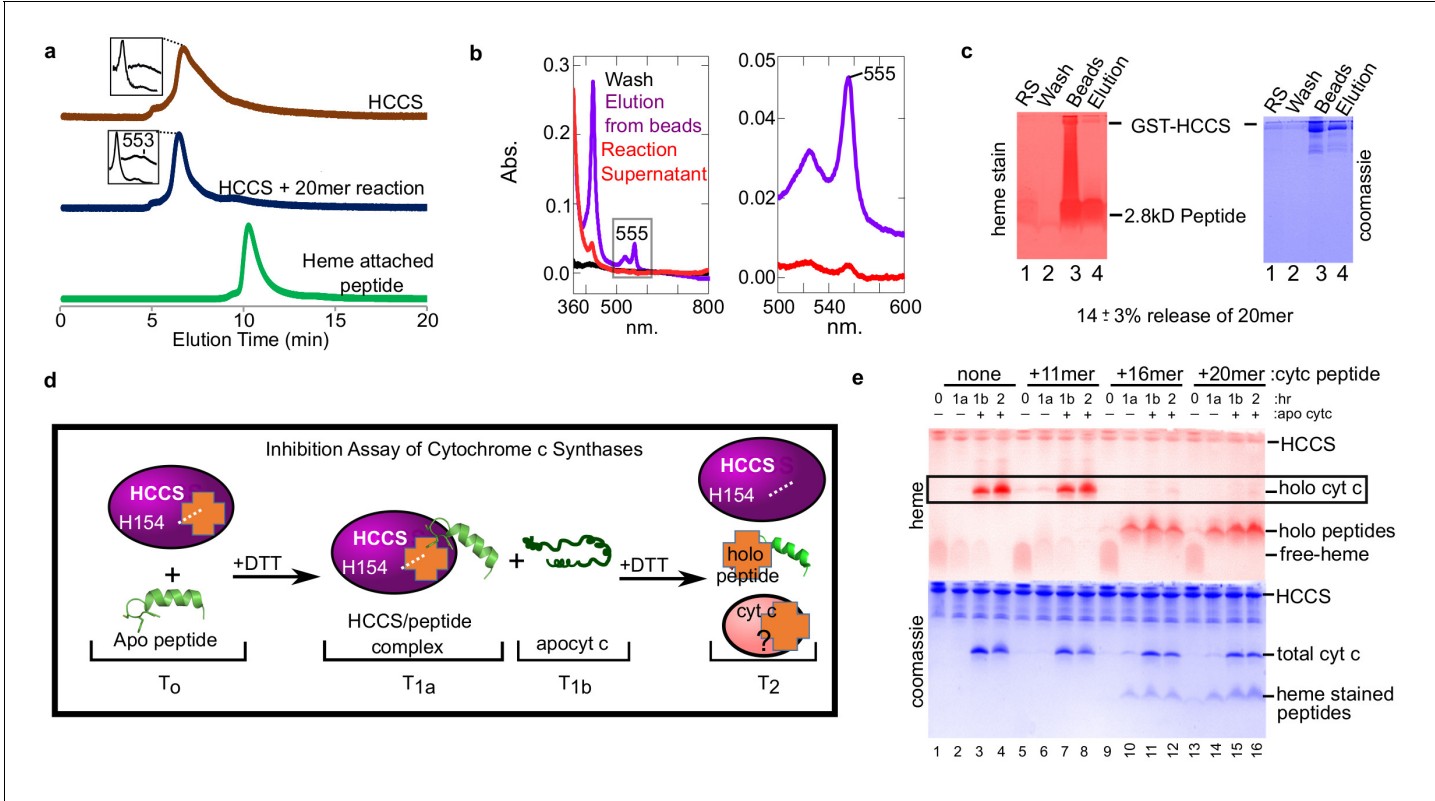

**Figure 4.** Peptides not released by HCCS can inhibit HCCS in vitro biosynthesis of cyt c. (**a**) The 20mer reaction was analyzed by SEC-HPLC (blue) and compared to HCCS alone (brown). The 'heme attached peptide' serves as a positive control for a released peptide (green). It is commercially available MP-11 (Sigma), an 11mer with heme attached that is purified from trypsinized cyt c. Insets show the spectra of the respective peaks. (**b**) The 'released product assay' (see *Figure 2a*) was performed with HCCS and the 20mer peptide. Glutathione eluted beads had a heme signal of 555 nm (purple) indicating a complex of HCCS with 20mer. The supernatant has little heme signal (red). (**c**) Tris–Tricine SDS–PAGE of the reaction supernatant (lane 1) and the elution from the beads (lane 4) shows that 14 ± 3% of the 20mer is released from HCCS. (**b** and **c**) are representative of seven trials and the estimated release is based on all trials. The standard deviation is provided. (**d**) Schematic of peptide inhibition assay with HCCS. $T_0$ – The in vitro reaction components HCCS and peptide are combined under anaerobic conditions, $T_{1a}$ – Addition of DTT initiates the reaction. Reaction incubates for 1 hr at 37 C, then the reaction is measured. $T_{1b}$– Apocyt c is added to the reaction to determine whether the peptide inhibits HCCS heme attachment to apocyt c. DTT is added to the reaction after $T_{1b}$ and incubated at 37 C for 1 hr. $T_2$– The final reaction products were analyzed by SDS–PAGE to determine if holocyt c was matured. (**e**) Reactions were separated by Tris–tricine SDS–PAGE and heme- and protein-stained. The 16 and 20mers inhibit HCCS maturation of apocyt c (lanes 11, 12, 15, 16). The 11mer or no peptide do not inhibit maturation of apocyt c (lanes 3, 4, 7, 8) (see boxed bands with holocyt c). The data is representative of three biological replicates.

HCCS reaction with the 56mer yields a 555 nm absorption (*Figure 3—figure supplement 1a*), it is likely not released.

To confirm that heme-attached peptides remain bound to HCCS, we used both HPLC SEC and the bead release assay described above (*Figure 2a*). HPLC separation (*Figure 4a*) showed that HCCS with the 20mer reaction (blue profile) eluted at the same time as HCCS alone (brown profile), not unexpected since a small 2.8 kD unreleased product would not significantly alter size exclusion properties. However, the spectra of the 20mer reaction from the HPLC SEC shows the signature of a HCCS-bound cyt c product, with a peak at 553 nm. This supports the conclusion that the heme attached 20mer remains bound to HCCS upon HPLC SEC, explaining why no heme-peptide product elutes separately (*Figure 4a*, compare blue and green profiles). Results of the bead release assay also show there is very little release of the heme-attached peptides from HCCS. Spectra of the reaction supernatant (red) exhibits very little heme (*Figure 4b*), unlike with full cyt c (*Figure 2b*, red). However, eluted HCCS from the beads show a spectra consistent with heme-attached peptide still bound, with a 555 nm peak (*Figure 4b*, purple). Quantitation of the level of heme-attached 20mer released from HCCS was carried out using the bead release assay (*Figure 4c*), with 14 ± 3% of the heme-attached peptide released from HCCS.

## Synthetic peptides as inhibitors of cyt c synthase activity

We evaluated whether peptides recognized by HCCS would act as inhibitors of heme attachment to subsequent addition of apocyt c. We carried out reactions with the three peptides for 1 hr, then added apocyt c, taking samples throughout (*Figure 4d*). The 11mer behaved as expected, as if no other substrate was present, with synthesis of cyt c occurring (in *Figure 4e*, compare lanes 1–4 and 5–8 boxed bands). This also suggests that the 11mer is not recognized by HCCS, in that it does not

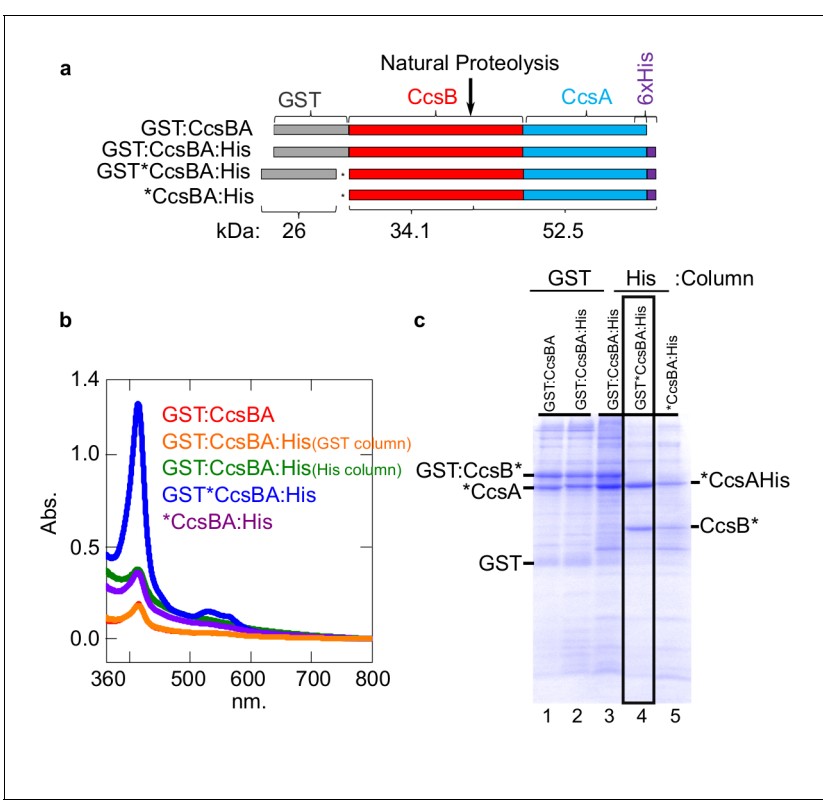

**Figure 5.** Construction of CcsBA with a C-terminal 6XHis affinity tag. (a) Schematics of CcsBA constructs used for overexpression and affinity purification. Gray, GST ORF; red, ccsB; blue ccsA; purple, C-terminal 6XHis tag. Site of natural proteolysis is shown with expected molecular weights of polypeptides. *Insertion of a stop/RBS/start cassette. UV–vis spectra Soret (~412 nm) is used to determine relative heme levels of 50 μg of purified CcsBA protein from the indicated construct. Spectra are representative of three independent purifications. (b) Affinity purifications of constructs in a. Affinity tag used for purification and relevant polypeptides are labeled. Boxed lane four is the His-tagged CcsBA used for these studies (except in *Figure 6b*). Data is representative of three biological replicates.

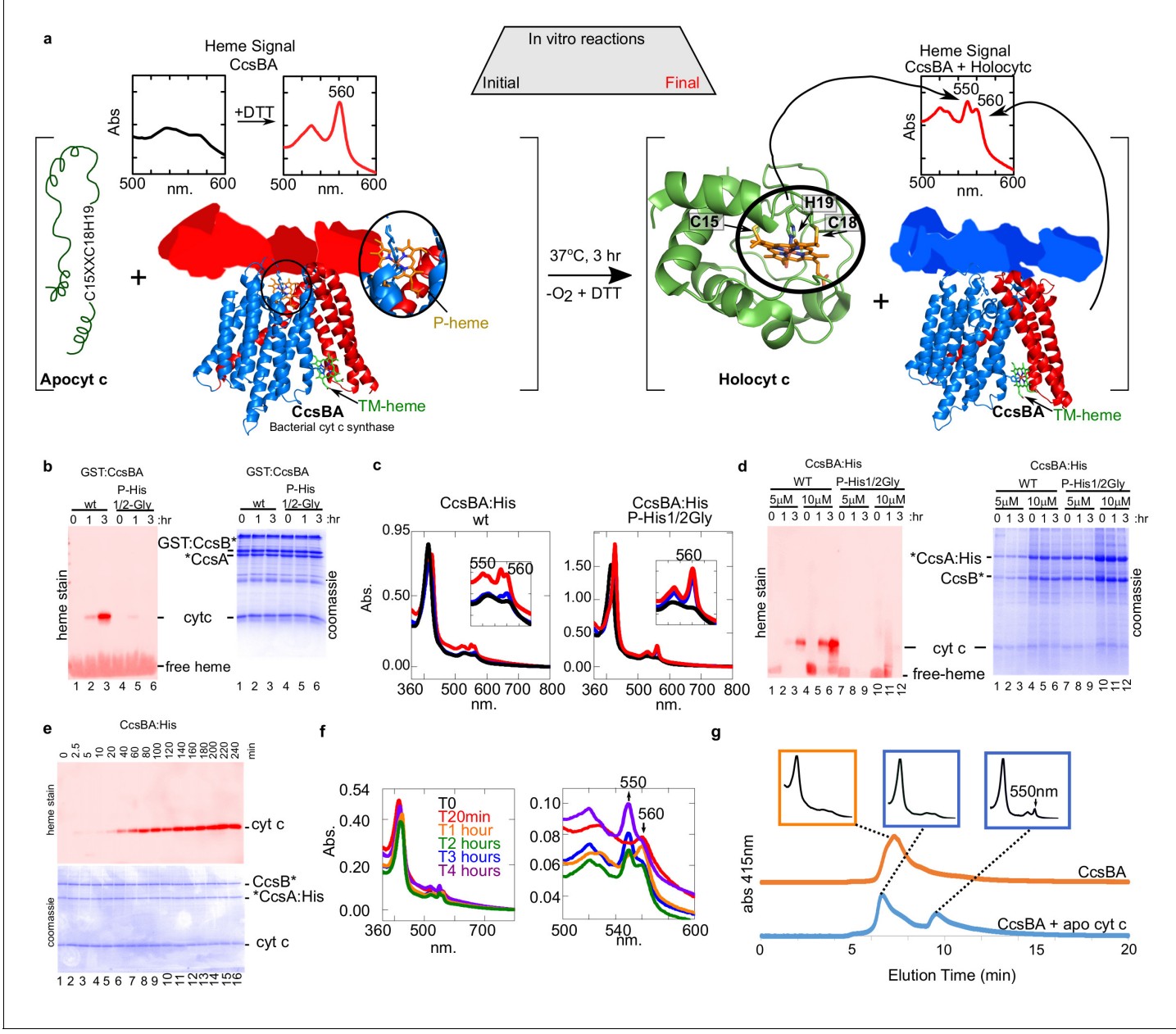

**Figure 6.** In vitro biosynthesis of cyt c by the bacterial synthase, CcsBA. (a) Schematic of the heme attachment reaction of CcsBA with apocyt c. Note, CcsBA has two heme binding sites, one in the periplasmic WWD domain (P-heme site, orange heme) and one in the transmembrane region (TM-heme site, green heme). CcsBA is proposed to traffick heme from the TM-heme site to the P-heme site for attachment to apocyt c. CcsBA model was generated by docking the TM-region (*Sutherland et al., 2018b*), with a cartoon of the periplasmic region. CcsBA is combined with apocyt c under anaerobic conditions and the reaction is initiated with DTT. UV–vis spectra (insets) show the initial reaction before (black) and after addition of DTT (red). After 3 hr, reaction products are holocyt c and monoheme CcsBA. (b) In vitro reaction with WT and P-His1/2Gly GST:CcsBA (5 µM) and apocyt c (20 µM). Samples were analyzed at 0, 1, 3 hr post-addition of DTT, separated by SDS–PAGE and maturation of holocyt c monitored by heme stain. CcsBA P-His1/2Gly is defective for heme binding in the P-heme site. (c) In vitro reaction with CcsBA:His (5 or 10 µM) and apocyt c (20 µM). Black, initial spectra; blue, 1 hr; red, 3 hr; 550 nm peak indicative of holocyt c; inset shows magnification of the $\beta/\alpha$ region. (d) Samples from (c) were analyzed at 0, 1, 3 hr post-DTT addition and analyzed as in (b, e), time course of in vitro reaction with CcsBA:His (5 µM) and apocytc (20 µM). Samples were taken at indicated timepoints and analyzed as in (b, f), UV–vis spectra of selected timepoints from (e), 550 nm peak indicative of holocyt c. Magnification of the $\beta/\alpha$ region is shown. (g), HPLC SEC separation of CcsBA (orange) and an in vitro reaction (blue). Monitored at 412 nm to detect heme. Insets show full spectra of indicated fractions.

prevent apocyt c from binding. However, both the 16mer (*Figure 4e*, lanes 9–12) and 20mer (lanes 13–16) showed heme attached to the peptides, but not to the apocyt c. We consider this inhibition of cyt c biogenesis (see Discussion). We conclude that alpha helix 1 is necessary and sufficient for recognition and attachment to the adjacent CXXCH motif. Our findings suggest that folding of cyt c is required for optimal release from the HCCS active site (see Discussion).

## In vitro reconstitution of CcsBA using apocyt C as substrate

Our previous studies with CcsBA have used recombinant GST-tagged CcsBA (from Helicobacter), shown to be functional in vivo and co-purify with endogenous heme (*Feissner et al., 2006*; *Frawley and Kranz, 2009*; *Sutherland et al., 2018b*). We concluded that CcsBA is both a heme exporter and a cyt c synthase with two heme binding sites (*Figure 6a*). To increase CcsBA yields for in vitro and future structural studies, we explored various tagging and expression strategies, ultimately selecting a C-terminal hexahistidine tagged CcsBA which gave high yields (*Figure 5a*). For unknown reasons, yields were higher when the GST ORF (with stop codon), as well as a new ribosome binding site upstream of *ccsBA* were used (threefold higher than GST-tagged or without the GST gene: *Figure 5a,b*). The purified hexahistidine tagged CcsBA still possessed the natural proteolysis site we have previously characterized (*Frawley and Kranz, 2009*; *Sutherland et al., 2018b*), resulting in two polypeptides (*Figure 5c*, lane 4, boxed). The GST*CcsBA:His construct is hereafter referred to as CcsBA:His. Using the anaerobic in vitro reconstitution conditions described above for HCCS, both the purified GST-CcsBA and metal-affinity purified CcsBA:His, both with endogenous heme, were active for heme attachment to apocyt c in vitro (*Figure 6a–d*). For further studies here, we used the CcsBA:His due to its higher yields. We have previously shown that while wt CcsBA has heme in both the P-His/WWD and TM-His sites (*Figure 6a*), the P-His variants possess heme only in the TM-His site (*Sutherland et al., 2018b*). GST:CcsBA P-His mutants are unable to attach heme in vivo to cyt c4, yet co-purified with heme (*Sutherland et al., 2018b*). Since heme is proposed to attach to apocyt c from the P-His/WWD site (*Figure 6a*), we tested whether the P-His variant functions in vitro, representing ideal negative controls for genuine in vitro attachments. Importantly, the GST:CcsBA P-His variant did not attach heme to apocyt c in vitro (*Figure 6b*). In vitro reactions with the wt CcsBA:His shows initial spectral signatures of *b*-heme (*Figure 6c*, black spectra). Within 1–3 hr, the wt CcsBA shows two peaks of reduced heme, one at 560 nm and a 550 nm peak that is characteristic of covalent heme attached in c-type cytochromes (*Figure 6c*, red spectra). It is likely that the *b*-heme (in the TM-His site) is responsible for the absorption remaining at 560 nm. These results were confirmed by SDS–PAGE and heme stains at the different time points (*Figure 6d*), confirming that the wt CcsBA formed cyt c. We conclude that purified wt CcsBA acts as a cyt c synthase in vitro and that heme is attached from the P-His/WWD domain, as hypothesized from in vivo results.

A time course of in vitro reactions with wt CcsBA shows that the covalent attachment to apocyt c is measurable at 20 min, reaching a maximum at approximately 3 hr (*Figure 6e*). Spectra at selected time points confirm these results (*Figure 6f*, see 550 nm formation). To determine whether cyt c is released from CcsBA and folds into its native state, we performed HPLC SEC on CcsBA alone and from a 3 hr reaction with apocyt c (*Figure 6g*). CcsBA in vitro synthesized cyt c is released and elutes at the same position as purified cyt c. The cyt c product (*Figure 6g*, last inset) is spectrally identical to cyt c produced by HCCS in vitro and to purified cyt c generated in vivo. We conclude that apocyt c is matured and released by CcsBA in vitro, with folding of cyt c into its native state.

## Peptide analogs of apocyt c are recognized by CcsBA and heme is covalently attached

Similar to HCCS studies, we used the 11, 16, and 20mer peptides (*Figure 3a*) and heme staining of tricine SDS–PAGE, to determine whether CcsBA attaches heme to peptide substrates and if so, what sequence or structural requirements are important. In CcsBA in vitro reactions, the 20mer, 16mer, and 11mer peptides each resulted in covalent heme after 3 hr in vitro reactions (*Figure 7a*). Spectral analyses also showed formation of 550 nm peaks (*Figure 7b*), including reactions with the 11mer, which was not recognized by HCCS. Because the 560 nm peak also remains in reactions, likely due to heme in the TM-His site, we used second-derivative spectra to delineate and quantitate the levels of attached heme present (*Figure 7b*, last panel, 550 nm), also confirming that all peptides possess the 550 nm absorption characteristic of c-type heme. The 56mer (alpha helix 1 and 2 of cyt

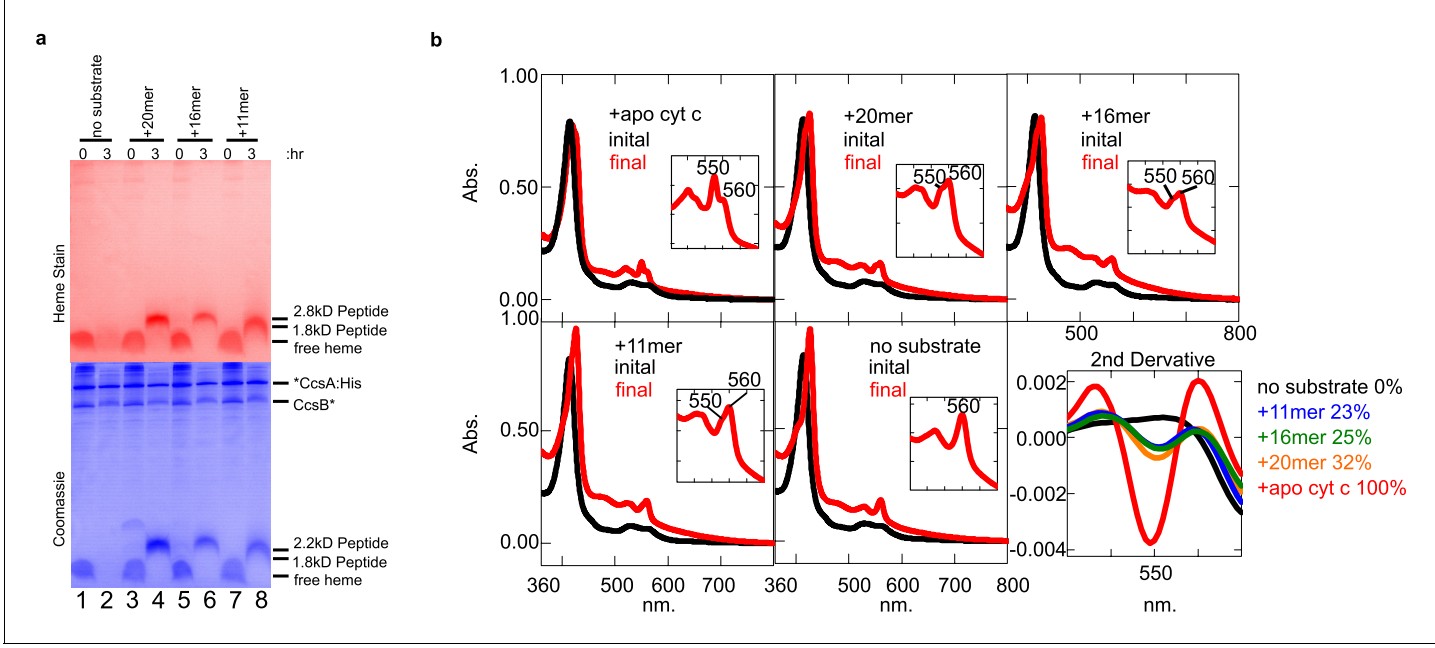

**Figure 7.** CcsBA recognition of peptides containing CXXCH for heme attachment. Peptides are described in *Figure 3a*. (a) Ten micromolar of CcsBA was incubated with 20 µM of the indicated peptide for 3 hr. Samples were taken at 0 and 3 hr and separated by Tris–tricine SDS–PAGE. Heme attached peptides were detected in lanes 4, 6, and 8 of the heme-stained gel (red). Total protein stain was completed after heme stain. Thus, coomassie stain shows signal from the heme, as well as total protein (blue). (b) UV–vis spectra of the samples in (a). Peaks at 550 nm are indicative of heme attached peptides while the peak at 560 nm reflects CcsBA-bound heme, possibly in the transmembrane domain Inset shows magnification of the β/α region. The second derivative of the spectra at 3 hr ('final') quantifies the signal at 550 nm, indicative of covalently attached heme. (a) and (b) are representative of three biological replicates.

The online version of this article includes the following source data and figure supplement(s) for figure 7:

**Figure supplement 1.** Attachment of heme to peptides by CcsBA.

**Figure supplement 1—source data 1.** Image J Pixel analysis of heme-stained bands.

**Figure supplement 2.** CcsBA and HCCS in vitro heme attachment to peptides.

**Figure supplement 3.** CcsBA anaerobic in vitro attachment of heme to CXXCH peptide variants.

c) and 9mer were also recognized and attached to heme by CcsBA (*Figure 7—figure supplement 1*). We conclude that the bacterial CcsBA cyt c synthase does not require the alpha helix 1 and that the recognition requirements are different than the mitochondrial HCCS (see *Figure 7—figure supplement 2* for parallel reactions of HCCS and CcsBA.)

## Recognition of peptide analogs with CXXCH cysteine and histidine substitutions, including non-natural thiol amino acids, by HCCS and CcsBA

The ability to biosynthesize heme-attached CXXCH peptides in vitro by HCCS and CcsBA facilitated a more detailed analysis of the cysteines and histidine in the substrates. For example, cysteine substitutions in the chemically synthesized peptides would determine whether each cysteine is required and whether non-standard thiol amino acids are recognized (*Table 1*). Homocysteine (HoC) has an additional carbon between the thiol and alpha carbon, while D-cysteine (D-C) rotates the thiol side-chain (see *Table 1* for structures). All substitutions were synthesized in the 20mer background since both HCCS and CcsBA could attach heme to it and the product is easily detected on heme stains of tricine SDS–PAGE. That is, if the peptide product has a covalent attachment, it will migrate at 2.8 kD and stain for heme (*Figure 3b*; *Figure 7a*; *Figure 7—figure supplement 2*). This method does not indicate whether a modified thiol (HoC or D-C) has a covalent attachment, so we also performed UV–vis and pyridine hemochrome spectroscopy to provide evidence of thioether formation. *Figure 3—figure supplement 2* and *Figure 7—figure supplement 3* show results of in vitro reactions of HCCS and CcsBA with the peptide analogs, as summarized in *Table 1*.

**Table 1.** Attachment of heme to peptides* by HCCS and CcsBA.

| Peptide name | Peptide sequence | HCCS Attachment | α-Peak (nm) | # cov attachments** | CcsBA Attachment | α-Peak (nm) | # cov attachments** |
|---|---|---|---|---|---|---|---|
| | | Minimal recognition primary/secondary sequences (lengths) | | | | | |
| HH Cyt c | …GDVEKGKKIFVQKCAQCHTVE… | Attached | 550 | 2 | Attached | 550, 560 | 2 |
| 56-mer | GDVEKGKKIFIMKCSQCHTVEKGGKHKTGPNLHGLFGRKTGQAPGYSYTAANKNKG | Attached | 555 | 2 | Attached | 550, 560 | 2 |
| 20 mer | GDVEKGKKIFIMKCSQCHTV | Attached | 553–554 | 2*** | Attached | 550, 560 | 2 |
| 16 mer | KGKKIFIMKCSQCHTV | Attached | 552 | 2 | Attached | 551, 560 | 2 |
| 11 mer | IMKCSQCHTVE | Not attached | 559 | 0 | Attached | 550, 560 | 2 |
| nine mer | KCSQCHTVE | Not attached | 559 | 0 | Attached | 550, 560 | 2 |
| | | Cysteine substitutions | | | | | |
| 20 mer Cys15S | GDVEKGKKIFIMKSSQCHTV | Attached | 555,559.5 | 1*** | Not attached | n.a. | 0 |
| 20 mer DCys15 | GDVEKGKKIFIMK(D-C)SQCHTV | Attached | 555 | 2*** | Not attached | 560 | 0 |
| 20 mer HoCys15 | GDVEKGKKIFIMKHoCSQCHTV | Attached | 555 | 1*** | Attached | 550, 560 | 2 |
| 20 mer Cys18S | GDVEKGKKIFIMKCSQSHTV | Attached | 559 | 1*** | Not attached | 560 | 0 |
| 20 mer DCys18 | GDVEKGKKIFIMKCSQ(D-C)HTV | Attached | 554 | 1*** | Not attached | 560 | 0 |
| 20 mer HoCys18 | GDVEKGKKIFIMKCSQHoCHTV | Attached | 558 | 1*** | Not attached | 560 | 0 |
| 20 mer Cys15S/Cys18S | GDVEKGKKIFIMKSSQSHTV | Not attached | 559 | 0 | Not attached | 560 | 0 |
| | | Histidine (of CXXCH) and lysine (K→D) substitutions for testing interaction models | | | | | |
| 20mer H19A | GDVEKGKKIFIMKCSQCATV | Not attached | 560 | 0 | Not attached | 560 | 0 |
| 20mer H19M | GDVEKGKKIFIMKCSQCMTV | Attached | 559 | 2 | Not attached | 560 | 0 |
| 20mer H19K | GDVEKGKKIFIMKCSQCKTV | Not attached | 560 | 0 | Not attached | 560 | 0 |
| 20mer K6A, K8D, K9D, K14D | GDVEAGDDIFIMDCSQCHTV | Not attached | 560 | 0 | Not attached | 560 | 0 |

*__Supplementary file 2__ contains additional information about peptides, eg. purity, synthesis co, and presence or absence of an N-terminal biotin-AHX tag.
**Number of covalent attachments determined by the final reaction spectra absorbance blue shifted from 560 nm and the presence of a heme stainable peptide.
***Pyridine hemochrome was performed to determine this number.

In the case of HCCS, all 20mer peptide variants possessed at least one covalent attachment with the exception of the SXXSH variant (__Table 1__, blue highlights). This indicates that HCCS does not require both cysteines for recognition, a conclusion consistent with in vivo results of engineered cyt

c substrate variants (*Babbitt et al., 2014*). Importantly, the HCCS/peptide complexes exhibit spectral signatures of purified HCCS/cyt c co-complex variants produced in vivo (*Babbitt et al., 2017*). For example, HCCS reactions with the SXXCH peptide shows a split alpha peak at 555/560 nm (*Figure 3—figure supplement 2b*), just as shown in vivo with the HCCS/Cys15Ser variant (*Babbitt et al., 2017*). Pyridine hemochrome spectra of HCCS reaction products were used to show whether the non-natural thiols were covalently attached. Both homocysteine and the DCys18 (*Figure 3—figure supplement 2f,g*), thiols were not attached, possessing only a single thioether, resulting in a hemochrome spectral peak of 552 nm that reflected attachment to Cys15. However, the DCys15 variant possessed two thioether attachments, thus both thiols reacted (*Figure 3—figure supplements 2c*, 550 nm pyridine hemochrome peak). This indicates that rotation of the first thiol (Cys 15) of the CXXCH substrate is more permissive at the active site of HCCS.

Lastly, we examined the role of the conserved H19 of the CXXCH motif. 20mer peptide were synthesized with H19M, H19A, and H19K substitutions. The H19A and H19K variants did not attach heme, while the H19M variant attached heme at low levels (*Figure 3—figure supplement 1*), suggesting methionine can act as a weak ligand in HCCS.

In the case of the bacterial CcsBA, an entirely different set of rules emerge for CXXCH substrate recognition (*Table 1*, compare blue to orange highlighted variants). Only one 20mer cysteine variant showed any covalent attachment: the first cysteine thiol replaced with a homocysteine (*Figure 7—figure supplement 3f*). The HoCys15 variant has two covalent linkages (550 nm peak), suggesting that the first thiol is more permissive in distance from the alpha carbon (i.e. of the first cysteine of CXXCH). Because DCys15 was not attached, unlike with HCCS, rotation of the first thiol may be less permissive at the CcsBA active site. No 20mers with histidine substitutions possess covalently attached heme with CcsBA (*Figure 7—figure supplement 1*).

## Discussion

It has been known for decades that the covalent, thioether attachment of heme in c-type cytochromes (to a CXXCH motif), requires accessory factors, including thioredoxins and cyt c synthases. A unique feature of cyt c biogenesis is that folding into its native structure occurs after cofactor (heme) attachment. Many elegant in vitro studies have concerned the folding of purified cyt c, typically after denaturation and renaturation to follow the folding pathway (e.g. *Hu et al., 2016*; *Pletneva et al., 2005*; *Yamada et al., 2013*). However, in vitro heme attachment by cyt c synthases has not been studied with purified components. Due in part to their membrane location, only recently have we been able to purify the detergent-solubilized synthases, mitochondrial HCCS (*San Francisco et al., 2013*) and bacterial CcsBA (*Frawley and Kranz, 2009*). CcsBA is an integral membrane protein that functions as a heme exporter and synthase, making its reconstitution particularly challenging. Here we have successfully reconstituted cyt c biogenesis with purified HCCS and CcsBA. Initially, we used apocyt c as substrate and endogenous heme that is co-purified with recombinant HCCS and CcsBA. For HCCS, we were also able to load heme into the active site, requiring His154, a process proposed as step one in biogenesis (*Figure 1—figure supplement 7*). Besides DTT for maintaining a reducing environment, no accessory factors other than HCCS and CcsBA are necessary. In vitro reactions result in stereochemical heme attachment, release of cyt c from the synthases, and proper folding into its native cyt c conformation. The cyt c possesses His19 (of CXXCH) and Met81 as axial ligands and its redox potential is identical to native cyt c purified from mitochondria (+253 mV).

In vitro reconstitution conditions (anaerobic, DTT) enabled the use of CXXCH containing peptides to study biogenesis and the substrate requirements for HCCS and CcsBA. In vitro reactions with HCCS and apocyt c proceed through all four steps (*Figure 1—figure supplement 7*), including step 4, release with cyt c folding. However, a 20mer peptide with CXXCH is very poorly released by HCCS, thus halting the process after step 3. In vivo we have demonstrated that single cysteine variants of cyt c (CXXCH motif) are released less than the wt cyt c, since more HCCS/cyt c complex and less cyt c product is purified (*Babbitt et al., 2014*). We proposed that thioether formation and consequent heme distortion contributes to release. Using cysteine peptide variants, we demonstrate in vitro that peptides with two thioethers release more than those with the single thioethers (*Figure 3—figure supplement 3*). Full cyt c is released at least 62 ± 5%, 20mer 14 ± 3%, and the SXXCH variant

5 ± 2% from HCCS. We conclude that folding of cyt c is necessary for optimal release from the HCCS active site (step 4).

For CcsBA, we have proposed that biogenesis involves heme trafficking from an internal membrane site, liganded by two TM-His residues, to an external domain called the WWD/P-His site (*Figure 6a*, *Frawley and Kranz, 2009*; *Sutherland et al., 2018b*). Subsequently, it is proposed that heme from the WWD/P-His site is stereochemically attached to apocyt c (CXXCH) (*Sutherland et al., 2018b*). Preliminary data on the spectral properties of peptides with heme attached by CcsBA appear to be released, unlike HCCS. Perhaps this release is mediated by the highly conserved WWD domain in the bacterial synthase, which interfaces with the edge of heme that faces the CXXCH substrate.

In vitro reconstitution with CXXCH peptides and analogs have shown that the substrate requirements for HCCS and CcsBA are quite different. There have been some in vivo studies that suggested that HCCS may require an N-terminally extended region (from CXXCH), yet such approaches do not rule out, for example, folding or stability issues (*Babbitt et al., 2016*; *Kleingardner and Bren, 2011*; *Zhang et al., 2014*). A direct, in vitro approach was needed. Here we synthesized multiple CXXCH peptides (*Figure 3a*): an 11mer lacking the N-terminal alpha helix 1 sequence, and a 16 and 20mer, which possess it. HCCS only recognizes and attaches heme to the 16 and 20mer but not the 11mer or a 9mer, while CcsBA attaches to all four peptides (*Table 1*, *Figure 3—figure supplement 1*). Structure of this alpha helix 1 sequence is predicted by PEP-FOLD (*Shen et al., 2014*; *Thévenet et al., 2012*) to form an alpha helix, consistent with experimental structure of cytochrome c. We conclude that the alpha helix 1 is a critical component recognized by HCCS, and that these peptides (16 and 20mers) present necessary and sufficient structures for recognition (*Figure 3a*). We used a Gremlin co-evolution/Rosetta approach (*Ovchinnikov et al., 2017*; *Ovchinnikov et al., 2015*) to determine the structure of HCCS, facilitated by almost a billion years of HCCS evolution (*Babbitt et al., 2015*). Heme was modeled into HCCS, constraining the His154 as an axial ligand, leaving the sixth ligand site open, likely bound to a weak ligand such as water (*Figure 8a*). *Figure 8b* displays the minimal 16mer substrate with heme. Heme binds to HCCS via His154 in step 1 (*Figure 1—figure supplement 7*), before binding of the 16mer substrate (step 2). The surface at the proposed active site of HCCS is acidic (*Figure 8a*), potentially interacting electrostatically with the basic features of alpha helix 1 (*Figure 8b*). Moreover, during step 2 of proposed model for HCCS function (*Figure 1—figure supplement 7*), His19 of apocyt c forms the second axial ligand to heme at the HCCS active site. In all peptides with alpha helix 1, spectral analysis indicated that His 19 formed this second axial ligand. We have confirmed the requirement for His19, testing three His19 variants of the 20mer peptide, H19M, H19A, and H19K (*Figure 3—figure supplement 1*). Only the H19M variant showed a low amount of attached heme, with a spectrum that also implies methionine can replace the weak ligand in HCCS (*Figure 3—figure supplement 1*). The minimal 16mer peptide, including the His19 ligand, is modeled into HCCS in *Figure 8c*. These models provide an initial structural basis for HCCS function, including testable predictions. For example, to test the electrostatic hypothesis, we changed all basic lysines to aspartates, retaining a predicted alpha helix 1 (*Figure 3—figure supplement 1*, *Table 1*). Heme was not attached to this peptide by HCCS, suggesting that the positive charge in alpha helix 1 is important.

For CcsBA, a limited sequence that includes CXXCH is necessary and sufficient. Results using peptide analogs with non-standard thiol amino acids are consistent with a more stringent requirement for the CXXCH motif for CcsBA. In this respect, because bacteria often recognize hundreds of c-type cytochromes (i.e. CXXCH motifs) it makes evolutionary sense to recognize only the CXXCH motif, than to have a more demanding three-dimensional structure.

We investigated the importance of the two thiols in CXXCH for recognition and thioether formation by synthesizing peptide analogs containing cysteine substitutions. Since the 20mer had heme attached by both HCCS and CcsBA, we used it as the base sequence for cysteine substitutions, as summarized in *Table 1*. Serine, homocysteine, and D-cysteine were substituted for each cysteine (of CXXCH). All substitutions were recognized by HCCS, having at least a single thioether, a result we attribute to the extended recognition requirement (alpha helix 1) and the His19 axial ligand (of CXXCH). We propose that this allows less dependency on the CXXCH motif. In contrast, CcsBA only recognized and attached heme to the variant with the first cysteine substituted by homocysteine. We propose that this is consistent with a more demanding recognition of the CXXCH motif at the active site of CcsBA. Clearly the serine substitutions cannot form thioethers. Consistent with this, for

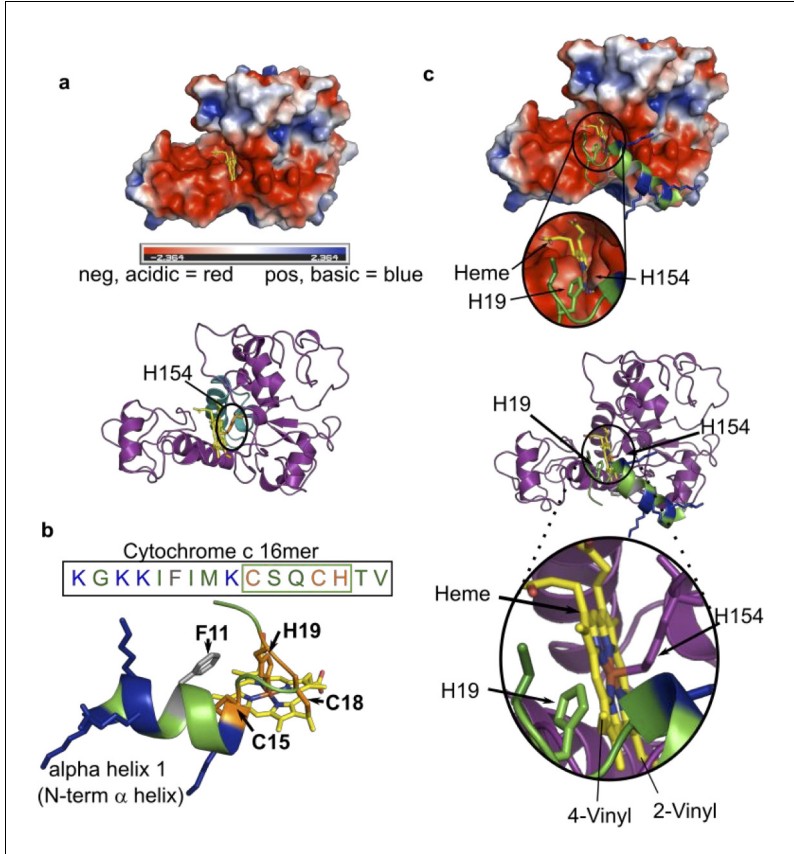

**Figure 8.** Modeled structure of HCCS using GREMLIN/Rosetta approach. (**a**) Electrostatic view of HCCS structure. Acidic surfaces are red; basic surfaces are blue. Heme is modeled with H154 ligand, within a pocket surrounded by acidic residues. Ribbon diagram of HCCS structure with domain II shown in teal and heme in yellow. The axial heme ligand H154 (orange) is shown. (**b**) Sequence and structure of the 16mer peptide substrate from cytochrome c; structure derived from PBD 3ZCF. Heme is shown in yellow, thioether bonds to Cys15 and Cys18 are indicated in orange with His19 serving as the axial ligand to heme. Positively charged (basic) residues (K) are blue. (**c**) Binding of the 16mer (or apocyt c), (step 2 of the four-step model) is displayed whereby H19 of cyt c ligands heme and positions cysteines for thioether attachment. (Top) Electrostatic view and interactions of HCCS structure and 16mer cyt c peptide with key interactions with HCCS and heme (yellow) and the 16mer peptide (green). (Bottom) A ribbon diagram with key interactions between HCCS H154 (purple) and heme (yellow), as well as formation of holo16mer peptide (green).

HCCS only the remaining cysteine had a thioether bond to heme. For HCCS, the only thiol amino acid analog with a thioether was D-cysteine substituted for the first cysteine (20mer DCys15 in *Table 1*). This suggests some rotational flexibility at the first thiol, but no 'vertical' flexibility since the homocysteine at Cys15 did not form a thioether. In contrast, for CcsBA, since only the homocysteine at Cys15 was attached, it may possess less rotational flexibility but more 'vertical' flexibility at the first cysteine at its active site. It is remarkable that in spite of the commonly proposed universal CXXCH motif for all c-type cytochromes, the bacterial and mitochondrial cyt c synthases have evolved quite different recognition determinants and thus, mechanisms. As discussed above, this is likely due to the limited c-type cyts in mitochondria (i.e. cyt c/cyt c1) but the large repertoire of c-type cyts in bacteria, each possessing CXXCH, and sometimes dozens of CXXCH motifs in a single bacterial protein.

Multiple approaches were used to demonstrate that CXXCH peptides with alpha helix one are not released by HCCS, with single cysteine substitutions even more tightly bound (see also *Figure 3—figure supplement 3*). Evidence is presented that peptides recognized by HCCS inhibit heme attachment to subsequently added cyt c. Thus, peptides are inhibitors. The basis for such inhibition will require more investigations, but two possible mechanisms are noted here. First, the

peptides specifically use the heme at the HCCS active site, thus precluding use by cyt c. Such a mechanism of inhibition might be considered specific dead-end use of a substrate. Second, in principle, tightly bound peptides that are not released may inhibit subsequent binding of new heme and cyt c substrates; thus, they act as substrate analog type inhibitors. Future studies will further explore these possibilities with both the mitochondrial HCCS and bacterial cyt c synthases.

# Materials and methods

## Key resources table

| Reagent type (species) or resource | Designation | Source or reference | Identifiers | Additional information |
|---|---|---|---|---|
| Strain, strain background (*Escherichia coli*) | NEB 5-α | New England Biolabs | fhuA2 Δ(argF-lacZ)U169 phoA glnV44 Φ80 Δ(lacZ)M15 gyrA96 recA1 relA1 endA1 thi-1 hsdR17 | Electrocompetent cells |
| Strain, strain background (*E. coli*) | C43(DE3) | doi:10.1006/jmbi.1996.0399; *Miroux and Walker, 1996* | F – ompT hsdSB (rB-mB-) gal dcm (DE3) | Electrocompetent cells |
| Strain, strain background (*E. coli*) | RK103 | doi:10.1111/j.1365–2958.2006.05132.x | MG1655 Δ*ccm::kan^R*, deleted for all *ccm* genes | Electrocompetent cells, protein expression, functional assays |
| Strain, strain background (*E. coli*) | MS36 | doi:10.1128/mBio.02134–18 | C43 Δ*ccm::kan^R*, deleted for all *ccm* genes | Electrocompetent cells, protein expression, functional assays |
| Antibody | Anti-equine heart cytochrome c (Rabbit polyclonal) | doi:10.1074/jbc.M116.741231 | | (1:10,000) |
| Recombinant DNA reagent | pRGK332 (plasmid) | doi:10.1111/j.1365–2958.2006.05132.x | pBAD *Bordetella pertussis* cytochrome c4:His | |
| Recombinant DNA reagent | pRGK368 (plasmid) | doi:10.1128/JB.01388–06; *Richard-Fogal et al., 2007* | pGEX *Helicobacter hepaticus* GST:CcsBA | |
| Recombinant DNA reagent | pRGK403 (plasmid) | doi:10.1073/pnas.1213897109 | pGEX GST:HCCS | |
| Recombinant DNA reagent | pRGK420 (plasmid) | doi:10.1073/pnas.1213897109 | pGEX GST:HCCS H154A | |
| Recombinant DNA reagent | pMCS97 (plasmid) | This study | pGEX *H. hepaticus* GST:CcsBA:His | See Materials and Methods and *Supplementary file 1* |
| Recombinant DNA reagent | pMCS64 (plasmid) | This study | pGEX *H. hepaticus* GST*CcsBA:His | See Materials and Methods and *Supplementary file 1* |
| Recombinant DNA reagent | pMCS154 (plasmid) | doi:10.1128/mBio.02134–18 | pGEX *H. hepaticus* GST:CcsBA:His | |
| Recombinant DNA reagent | pMCS558 (plasmid) | This study | pGEX *H. hepaticus* *CcsBA:His | See Materials and Methods and *Supplementary file 1* |
| Recombinant DNA reagent | MCS598 (plasmid) | This study | pGEX *H. hepaticus* GST*CcsBA:His P-His1/2G | See Materials and Methods and *Supplementary file 1* |
| Sequence-based reagent | pGEX GST*F | This study | PCR Primer | tcggatctggttccgcgttgaag gaggaaggatccatgatgaat |
| Sequence-based reagent | pGEX GST*R | This study | PCR Primer | attcatcatggatccttcctccttca acgcggaaccagatccga |
| Sequence-based reagent | pGEX CcsBA 6HisF | This study | PCR Primer | gagtgcttgatatgccccatttaca tcaccatcaccatcactaactcgagcggc |

*Continued on next page*

*Continued*

| Reagent type (species) or resource | Designation | Source or reference | Identifiers | Additional information |
|---|---|---|---|---|
| Sequence-based reagent | pGEX CcsBA 6HisR | This study | PCR Primer | gccgctcgagttagtgatggtgatggt gatgtaaatggggcatatcaagcactc |
| Sequence-based reagent | MSP5 | This study | PCR Primer | gtgcttaaatcttattggctcaa cattggcgtctccgtcatca |
| Sequence-based reagent | MSP6 | This study | PCR Primer | tgatgacggagacgccaatgt tgagccaataagatttaagcac |
| Sequence-based reagent | MSP7 | This study | PCR Primer | ttattatctcacaggtatgggc agctatgccgcaggagaa |
| Sequence-based reagent | MSP8 | This study | PCR Primer | ttctcctgcggcatagctgcc catacctgtgagataataa |
| Peptide, recombinant protein | Holo-MP11 | Sigma-Aldrich | Cat. #M6756 | See *Supplementary file 2* |
| Peptide, recombinant protein | Biotin-56-mer | RS-synthesis | | See *Supplementary file 2* |
| Peptide, recombinant protein | Biotin-20 mer | RS-synthesis | | See *Supplementary file 2* |
| Peptide, recombinant protein | 20-mer | RS-synthesis | | See *Supplementary file 2* |
| Peptide, recombinant protein | Biotin-16 mer | RS-synthesis | | See *Supplementary file 2* |
| Peptide, recombinant protein | 11 mer | RS-synthesis | | See *Supplementary file 2* |
| Peptide, recombinant protein | nine mer-biotin | RS-synthesis | | See *Supplementary file 2* |
| Peptide, recombinant protein | 20 mer Cys15S | RS-synthesis | | See *Supplementary file 2* |
| Peptide, recombinant protein | 20 mer DCys15 | CS Bio Co | | See *Supplementary file 2* |
| Peptide, recombinant protein | 20 mer HoCys15 | CS Bio Co | | See *Supplementary file 2* |
| Peptide, recombinant protein | 20 mer Cys18S | CS Bio Co | | See *Supplementary file 2* |
| Peptide, recombinant protein | 20 mer DCys18 | CS Bio Co | | See *Supplementary file 2* |
| Peptide, recombinant protein | 20 mer HoCys18 | CS Bio Co | | See *Supplementary file 2* |
| Peptide, recombinant protein | 20 mer Cys15S/Cys18S | CS Bio Co | | See *Supplementary file 2* |
| Peptide, recombinant protein | 20mer H19A | RS-synthesis | | See *Supplementary file 2* |
| Peptide, recombinant protein | 20mer H19M | RS-synthesis | | See *Supplementary file 2* |
| Peptide, recombinant protein | 20mer H19K | RS-synthesis | | See *Supplementary file 2* |
| Peptide, recombinant protein | 20mer K6A, K8D, K9D, K14D | RS-synthesis | | See *Supplementary file 2* |
| Peptide, recombinant protein | Pierce-glutathione agarose | Thermo Scientific | Cat. #16101 | |
| Peptide, recombinant protein | Talon Resin | TaKaRa | Cat. #635503 | |
| Chemical compound, drug | Hematin | Fisher | Cat. #AAA1851803 | |

*Continued on next page*

*Continued*

| Reagent type (species) or resource | Designation | Source or reference | Identifiers | Additional information |
|---|---|---|---|---|
| Chemical compound, drug | *N, N, N', N'*-tetramethylbenzidine (TMBZ) | Sigma | Cat. #1086220001 | |
| Chemical compound, drug | Equine horse-heart cytochrome *c* | Sigma | Cat. #C2506 | |
| Chemical compound, drug | 2,6-dichloroindophenolate hydrate (DCPIP) | Sigma | Cat. #D-1878 | |
| Commercial assay or kit | Pierce-SuperSignal West Femto ECL reagent | Thermo Scientific | Cat. #PI34096 | |

## Bacterial growth conditions

*Escherichia coli* strains were grown in Luria-Bertani (LB; Difco) broth with selective antibiotics and inducing reagents as required. Antibiotic/induction concentrations: carbenicillin, 50 µg/ml; chloramphenicol, 20 µg/ml, isopropyl $\beta$-D-1-thiogalactopyranoside (IPTG; Gold Biotechnology), 1.0 mM or 0.1 mM; arabinose (alfa Aesar), 0.2% (wt/vol).

## Construction of strains and plasmids

Cloning was performed using *E. coli* NEB-5 $\alpha$ with the QuikChange II site-directed mutagenesis kit (Agilent Technologies) following the manufacturer's instructions. Strains, plasmid, and primer lists are provided in *Supplementary file 1* in supplemental material.

## Protein purifications

GST-HCCS purifications were performed as previously described (*San Francisco et al., 2013*). Briefly, starter cultures (100 ml) were grown overnight at 37°C and 200 rpm. Starter cultures were used to inoculate 1 l of LB supplemented with appropriate antibiotics. One liter cultures were grown at 37°C and 120 rpm for 1 hr, and next expression of GST-HCCS was induced with 0.1 mM IPTG. Cells were harvested after 5 hr by centrifugation at 4500 g and cell pellets were stored at −80°C. Cell pellets were resuspended in PBS supplemented with 1 mM phenylmethansulfonul fluoride (PMSF), lysed by sonication (Branson250 sonicator), and cleared of cell debris by centrifugation at 24,000 g for 30 min at 4°C. Separation of soluble and membrane fractions was achieved by high-speed ultracentrifugation at 100,000 g for 45 min at 4°C. Membrane pellets were solubilized in 50 mM Tris pH 8, 150 mM NaCl, and 1% Triton X-100 for 1 hr on ice. Solubilized membranes were added to glutathione agarose (Pierce) for batch pulldown. Note, GST-HCCS used for in vitro reactions were heme loaded at this step by addition of 5 µM hemin during batch pulldown (see below). Columns were washed by gravity flow and eluted in 50 mM Tris pH8, 150 mM NaCl, and 0.02% Triton X-100 supplemented with 20 mM glutathione. Elution was concentrated using Amicon Ultra Centrifugal Filters (Millipore), and protein concentration was determined by Bradford assay (Sigma).

GST-CcsBA and GST-CcsBA:His purifications were performed as previously described (*Frawley and Kranz, 2009*; *Sutherland et al., 2018b*). Briefly, 5 ml starter cultures were grown for ~8 hr at 37°C with rocking. Starter cultures were diluted 1:200 into 1 l LB with selective antibiotics and grown overnight at 24°C and 240 rpm to saturation. Expression of GST-CcsBA was induced with 1 mM IPTG for 6 hr, cells were harvested at 4500 g, and cell pellets were stored at −80°C. Cell pellets were resuspended in Resin Buffer (20 mM Tris pH8, 100 mM NaCl) supplemented with 1 mM PMSF and 1 mg/ml egg white lysozyme (Sigma-Aldrich). Cells were lysed, cleared of debris, and separation of membrane fraction was performed as described for GST-HCCS above. Membrane pellets were solubilized in Resin Buffer with 1% n-dodecyl-β-d-maltopyranoside (DDM; Anatrace) and batch purified for 2 hr with glutathione agarose (Pierce). Columns were washed by gravity flow using Resin Buffer with 0.02% DDM and eluted in Resin Buffer with 0.02% DDM and 20 mM glutathione. Elution

was concentrated using Amicon Ultra Centrifugal Filters (Millipore), and protein concentration was determined by Bradford assay (Sigma).

GST-CcsBA:His, GST*CcsBA:His and *CcsBA:His were performed as described above for GST-CcsBA with the following modifications. Batch pulldowns were performed using Talon Affinity Metal Resin (Takara). Gravity flow washes were performed in Resin buffer with 0.02% DDM supplemented with 0 mM imidazole (wash 1), 2 mM imidazole (wash 2), and 5 mM imidazole (wash 3). Protein was eluted in Resin Buffer with 0.02% DDM and 125 mM imidazole. Elution was concentrated using Amicon Ultra Centrifugal Filters (Millipore), and protein concentration was determined by Bradford assay (Sigma).

## Heme loading of HCCS

To increase heme co-purification of GST-HCCS, exogenous heme was added to the affinity purification, resulting in 'heme loaded' HCCS. During binding of the solubilized membrane preparations to glutathione agarose, hemin (1.3 mg/ml in DMSO) was added to a final concentration of 5 µM. Heme loading increases HCCS heme co-purification from ~10% to ~30%. To determine the optimal concentration of hemin, a range of values was tested (see *Figure 1—figure supplement 2*). After batch affinity purification, the column was washed (removing unbound hemin) and eluted as described in protein purification section.

## Heme staining, SYPRO Ruby, and Coomassie protein staining and immunoblotting

Samples were prepared in loading dye at 1:1 (v/v) that did not contain reducing agents and were not boiled to maintain heme signals. Samples were separated by SDS–PAGE or Tricine SDS–PAGE (peptides). Heme staining was performed by transfer to nitrocellulose and detection of heme signal using the SuperSignal Femto kit (Pierce) (*Feissner et al., 2003*), with imaging on a LI-COR odyssey Fc (LI-Cor Biosciences) or by in-gel heme stain with *N, N, N', N'*-tetramethylbenzidine (TMBZ) (*Feissner et al., 2003*; *Francis and Becker, 1984*; *Thomas et al., 1976*). Total protein was detected by staining SDS–PAGE gels with Coomassie stain or nitrocellulose blots with SYPRO Ruby Blot Stain according to the manufacturer's instructions (Molecular Probes). Immunoblots using an antibody specific to equine heart cytochrome *c* (Cocalico Biologics) were performed as previously described (*Babbitt et al., 2016*).

## UV–vis absorption spectroscopy

UV–vis absorption spectroscopy was obtained with a Shimadzu UV-1800 spectrophotometer. Spectra were recorded in the assay buffer and under aerobic or anaerobic conditions as indicated. Heme quantification by Soret absorbance was performed with 50 µg of protein. Pyridine hemochrome assays were performed as previously described (*Berry and Trumpower, 1987*) in the assay buffer. If needed, sodium dithionite powder was used for protein reduction. Maturation of peptides by CcsBA was assessed by measuring the maximum or minimum of the second derivative of the final reaction spectrum.

## In vitro reconstitution of synthase function

In vitro reconstitutions were performed aerobically (HCCS) or anaerobically (HCCS and CcsBA). For anaerobic reactions, all reagents were equilibrated with $N_2$ (95%) and $H_2$ (5%) in a Coy anaerobic airlock chamber. Affinity purified synthase (HCCS or CcsBA) was combined with apo equine heart cytochrome c or apo peptide at indicated concentrations. Apo cyt c and peptide concentrations used were determined to be within the range for maximal heme attachment as determined by a titration. An initial spectra and sample for SDS–PAGE analysis were obtained. Five millimolar DTT was added to initiate the reaction. Reactions were placed at 37°C, and spectra and gel samples were taken at indicated time points. Gel samples were immediately placed in loading dye (1:1 v/v) to stop the reaction.

## Apo equine heart cytochrome c preparation

Apocytochrome c preparation was modified from *Babul and Stellwagen, 1972*. Cytochrome c from equine (horse) heart was obtained from Sigma, and a 1 ml 10 mg/ml solution was prepared in water.

To remove heme, 200 µl of glacial acetic acid and 1.5 ml of 0.8% silver sulfate were added and the solution was incubated at 44°C for 4 hr. Sample was dialyzed in 0.2 M acetic acid overnight at 4°C. To precipitate apo cytochrome c and remove silver, sample was transferred to a conical tube and 10 volumes of cold acid acetone were added. Apo cytochrome c was pelleted by spinning at 15,000 rpm for 20 min at 4°C. The pellet was washed with acid acetone and pelleted three times. The apo-protein was resuspended in 0.2M acetic acid (~1 ml), and solid urea was added until the solution turned clear. A 25-fold molar excess of 2-mercaptoethanol was added and incubated at room temperature to remove silver sulfate. Apo cytochrome c was clarified by centrifugation at 12,000 rpm for 10 min at room temperature. Supernatant was dialyzed in 0.2 M acetic acid overnight and buffer exchanged into PBS by concentration in an Amicon Concentrator with 3 kDa molecular weight cut-off. Protein concentration was determined using a BSA standard curve and Coomassie protein staining on SDS–PAGE.

## High-performance liquid chromatography

Affinity purified proteins or indicated in vitro reactions were resolved on an Agilent 1100 HPLC system equipped with an Agilent SEC-3 column in the purification or in vitro reaction buffer.

## In vitro HCCS-tethered released product reaction

To determine whether in vitro synthesized cytochrome c (or peptide) was released from the synthase, GST-HCCS bound to glutathione agarose (75 µl) was combined with apocyt c (or peptide) under standard in vitro conditions (100 µl volume in addition to the 75 µl of beads). After a 1 hr reaction, the glutathione agarose-bound GST-HCCS were pelleted, and the supernatant was collected. Subsequently, the beads were washed to allow for analysis of protein retained on the beads. The bead fraction and supernatant were separated by SDS–PAGE and heme stained to determine which fraction contained heme attached cyt c. The amount of holo cyt c/peptide matured and released was quantitated using Image J (*Rasband, 1997*) by determining the ratio of the supernatant derived heme band with the total peptide band signal (beads plus supernatant). The supernatant was further analyzed by UV–vis spectroscopy.

## CD spectroscopy of released cytochrome c

The supernatant from the released product assay (above) was extracted and pooled in an anaerobic environment and then concentrated in a 3K VivaspinTurbo cutoff filter (Sartorius) to obtain 300 µl of 0.4 mg/ml cyt c as determined by heme absorbance at 550 nm (ext. coef. 29.5 mM$^{-1}$ cm$^{-1}$). The near UV (500–300 nm) signal was measured on a Jasco J-815 at room temperature in the reaction buffer (20 mM Tris pH 8.0, 100 mM NaCl, 0.02% DDM, 5 mM DTT). The machine sensitivity was 100 mdeg, the data pitch was 0.5 nm, the scanning mode was continuous, the scanning speed was 50 nm/min, the response rate was 1 s, the bandwidth was 1 nm, and five accumulations were taken (*Mendez et al., 2017*). A blank sample was subtracted. To compare the absorbance of the released assay product to human cyt c, each CD spectra was subtracted from a blank sample and divided by the absorbance of the protein in the CD machine. The samples were overlaid and coincide with each other.

## Determination of heme redox potential

Redox potential of in vitro synthesized equine heart cytochrome c was determined by a modified Massey method as described in *Sutherland et al., 2016*, with the following modifications: The absorbance change of heme was monitored at the alpha peak at 550 nm (negligible contribution from reference dye) and the reduction of the reference dye, dichlorophenolindophenol, at 636 nm (negligible contribution from heme).

## In vitro inhibition assay

To determine whether the CXXCH containing peptides inhibited maturation of cytochrome c (i.e. heme attachment), a two-step reaction was performed. Step 1: 10 µM GST-HCCS (30% heme occupancy) was combined with 10 µM apo peptide for a 1 hr in vitro reaction. After 1 hr, UV–vis spectra were performed, and a sample was collected for gel analysis. Step 2: 20 µM apo cytochrome c was added to the reaction. After 1 hr, UV–vis spectra were performed and a sample was collected for

gel analysis. To determine whether cytochrome c maturation was inhibited or not inhibited by the peptide, heme stain and Coomassie total protein stain were performed.

### In vivo functional (heme attachment) assays

Assays were performed as in *Sutherland et al., 2018b*. Detailed methods are provided in Supplemental Methods.

### Generation of the modeled structure of HCCS

The HCCS structure was produced using Rosetta, which was informed by structural motifs (Robetta) and coevolutionary data (Gremlin) as has been described (*Ovchinnikov et al., 2017*; *Ovchinnikov et al., 2015*; *Sutherland et al., 2018b*; *Sutherland et al., 2018a*), and will be detailed later.

## Acknowledgements

We thank Jen Hsu and Jeff Orf for technical assistance on the HCCS in vitro assay and Hani S Zaher and Kyusik Kim for use of their HPLC. This work was funded by the National Institutes of Health (R01 GM47909 to RGK).

## Additional information

### Competing interests

Deanna L Mendez, Robert G Kranz: Washington University (Robert Kranz and Deanna Mendez, inventors) has applied for a provisional patent titled "PEPTIDE-BASED INHIBITORS OF CYTOCHROME BIOGENESIS". APPLICATION NUMBER: 63158593. The other authors declare that no competing interests exist.

### Funding

| Funder | Grant reference number | Author |
| --- | --- | --- |
| National Institutes of Health | R01 GM47909 | Robert G Kranz |

The funders had no role in study design, data collection and interpretation, or the decision to submit the work for publication.

### Author contributions

Molly C Sutherland, Deanna L Mendez, Conceptualization, Investigation, Methodology, Writing - original draft, Writing - review and editing; Shalon E Babbitt, Conceptualization, Investigation, Methodology, Writing - review and editing; Dustin E Tillman, Olga Melnikov, Noah T Prizant, Andrea L Collier, Investigation, Writing - review and editing; Nathan L Tran, Investigation, Writing - review and editing, Carried out HCCS structural predictions; Robert G Kranz, Conceptualization, Supervision, Funding acquisition, Investigation, Methodology, Writing - original draft, Writing - review and editing

### Author ORCIDs

Molly C Sutherland https://orcid.org/0000-0002-7932-5339
Dustin E Tillman https://orcid.org/0000-0002-7450-0927
Nathan L Tran https://orcid.org/0000-0002-7917-3945
Robert G Kranz https://orcid.org/0000-0002-4309-9413

### Decision letter and Author response

Decision letter https://doi.org/10.7554/eLife.64891.sa1
Author response https://doi.org/10.7554/eLife.64891.sa2

## Additional files

### Supplementary files

- Supplementary file 1. Relevant strains, plasmids, and primers.
- Supplementary file 2. Apo peptides used for in vitro assays.
- Transparent reporting form

### Data availability

Data generated is provided in the manuscript. Source data files are provided for Figure 2C/Figure 4C/Figure 3—figure supplement 1C; Figure 3—figure supplement 1C; Figure 7—figure supplement 1C.

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
