## [Decision Letter]

**Acceptance summary:**

This manuscript provides the molecular mechanisms for how two seemingly disparate cytochrome c synthases, a human mitochondrial (HCCS) and bacterial (CcsBA), utilizes surprisingly similar steps for covalent attachment of heme to cytochrome c.

**Decision letter after peer review:**

Thank you for submitting your article "in vitro reconstitution reveals major differences between human and bacterial cytochrome c synthases" for consideration by *eLife*. Your article has been reviewed by 3 peer reviewers, including Iqbal Hamza as the Reviewing Editor and Reviewer #1, and the evaluation has been overseen by Michael Marletta as the Senior Editor. The following individual involved in review of your submission has agreed to reveal their identity: Amy E. Medlock (Reviewer #2).

Essential Revisions:

As you can see below, all three reviewers recommended publication of the manuscript. They all agreed that the work is thorough and rigorous. The parallel work with the human and bacterial CCS provides a broad and universal appeal. The recommendations by the reviewers should not result in any additional experiments, but the revised manuscript should provide better clarity and presentation, especially with the figure layout. We look forward to the revised manuscript.

*Reviewer #1:*

This is a thoughtful and elegant study of how human (HCCS) and bacterial (CcsBA) cytochrome c synthases provide heme and catalyze covalent attachment of heme to its cognate cytochrome c recipient. The study is comprehensive and is well-done. I have only a few suggestions to help improve the readability of this manuscript and its overall impact.

1. The authors have performed a comprehensive analyses using physicochemical and spectroscopic methods comparing their assay-generated holo cyt c to purified cyt c. Is there a functional assay to evaluate whether the holo cyt c generated in their assays are actually biologically active?

2. I had a difficult time seeing the figures and reading some of the figure legends (Figure 2 is especially poorly-written).

3. In Figure 2h, the authors first added the peptide (11, 16, and 20mers) before performing a competition with apocyt c to determine how well heme transfer versus release occurs with HCCS. Have the authors performed this experiment in the presence of increasing concentrations of apocytc? What will be effect of the peptides if apocyt c was added simultaneously with the peptide for "hemylation"?

*Reviewer #2:*

The goal of the authors was to construct an in vitro system for cytochrome c maturation in order to identify and compare the attachment elements necessary for the human and bacterial systems. This work is a large technical advance and thus allowed the characterization of the different cytochrome c biogenesis systems at a much finer level than was previously possible. The key finding in the manuscript are supported by the data and set the stage for future studies to evaluate the mechanisms of cytochrome maturation inhibition and the development of therapeutics to target cytochrome c.

The manuscript by Sutherland et al. entitled "in vitro reconstitution reveals major differences between human and bacterial cytochrome c synthases" describes the first in vitro systems for the reconstitution of cytochrome c by the human system (system III) and a bacterial system (system II). These in vitro systems revealed several important aspects of HCCS (system III) and CcsBA (system II). First, besides the reductant (DTT), both synthases can function in vitro without any other protein or protein cofactor. Second, they produce holo-cytochrome c that structurally, as evaluated by CD, and functionally, as evaluated by redox titrations, is identical to holoprotein produced in vivo. The development of the in vitro maturation of cytochrome c then allowed the detailed analysis of the necessary recognition sequence or elements for HCCS (system III) and CcsBA (system II) cytochrome c maturation. Synthetic peptides of varying lengths with the CXXCH motif were used to determine the minimal structural or sequence units necessary. Elements necessary for recognition between HCCS and CcsBA differed at both the amino acid and length or secondary structure levels. While HCCS required alpha helix 1 and only one cysteine in the CXXCH motif, CcsBA required both cysteines and the histidine of the CXXCH motif, but not helix 1. Interestingly these peptides inhibited cytochrome c maturation, in the case of HCCS by not being released from the enzyme. The development of this in vitro system, characterization of the minimal recognition sequences for HCCS and CcsBA, comparison of HCCS and CcsBA recognition motifs and inhibition of synthases by peptides are an important discovery and relevant for considering cytochrome c maturation as a target for antimicrobial compounds. The concerns outlined below should be easily addressed by the authors and would likely require minor edits to the text. Thus this work warrant publication in *eLife* with the following revisions.

1. A claim of the paper is that helix 1 is necessary, yet no experiments were done to determine if the helix forms in the peptides. Has the formation of the helix in the peptides, in which it is proposed to form, been investigated (such as by CD or crystallization)? If not, can the conclusion be modified to address this?

2. For the 16mer, it appears that only the biotin-16mer (Figure 2a and Table S2) was used for the studies whereas the Biotin-20mer and the 20mer are listed. Does the Biotin-20mer behave identically to the 20mer? Discussion and confirmation of this in the text is important to show that the biotin tag doesn't affect the peptide structure and thus the recognition.

3. For the CcsBA system the time necessary for in vitro cytochrome c biogenesis is 3 hours while the HCCs is 1 hour. Why the difference? Could the system be a missing factor? This could be addressed within the text.

4. Statistical information is missing. While quantification on a number of experiments was performed, no statistical analysis is included. One example is in the results lines 285 to 286 and discussion lines 484 to 487, the release of cytochrome c, the 20mer and the 20mer SXXCH are discussed and data presented in Figures 2e and S4. While multiple experiments were performed no statistical analysis was provided. This should be added to understand the accuracy of these measurements and support for the conclusion.

5. Based on the lack of release of peptides by HCCS it is concluded that the folding of the protein is necessary for optimal release. If helix 1 is formed and necessary for attachment, this suggests that some secondary structural elements are formed and that tertiary structure is necessary for release. As noted above investigation of the structural elements of the peptides would strengthen and this could be added to the text.

6. In the Results section there is no summary of the H and K mutants, it would be helpful to add one or two sentences as this is a significant part of the discussion and basis for the model presented in Figure 5.

*Reviewer #3:*

The manuscript has two main thrusts. Firstly, the authors developed an in vitro assay for cytochrome c biosynthesis, namely the covalent attachment of heme to apocytochrome c catalyzed by cytochrome c synthase. They reconstituted the synthases from both human (mitochrondrial) and bacterial enzymes, which are structurally unrelated proteins despite catalyzing the same reaction. Secondly, they use this reconstitution assay to probe features of cytochrome c synthesis by the two enzymes and show numerous differences in the mechanism by the two enzymes. A practical implication of this is the use of bacterial-specific peptide inhibitors for antibiotic therapy.

The reconstitution conditions are described thoroughly and are well-controlled. They show that the enzyme bound to glutathione beads behaves similarly to the free protein and, overall, that the in vitro assay recapitulates what occurs in cells.

Using peptides containing the CXXCH heme attachment site of cytochrome c as substrates for the human synthase (HCCS), the authors show that peptides require the alpha helix adjacent to the attachment site for heme attachment. However, the reaction product was not released from the enzyme. This suggests that cytochrome c folding is needed for product release. These peptides were able to inhibit heme attachment to cytochrome c by HCCS.

The bacterial cytochrome c synthase from Helicobacter (CcsBA) was also reconstituted in vitro. Bacterial cytochrome c biogenesis involves numerous accessory proteins, and therefore it is useful to know that CcsBA was sufficient to catalyze heme attachment. Unlike the human enzyme, it was able to attach heme to a peptide lacking the alpha helix, which correlates with the fact that the bacterial cytochrome c normally lacks the helix adjacent to the CXXCH binding site. Thus, the mitochondrial and bacterial cytochrome c synthases recognize different parts of the apo-cytochrome c substrate.

Differences between the two enzymes were also observed using peptides with thiol substitutions within the CXXCH site. These substitutions included serine, D-cysteine and homocysteine. The mitochrondrial enzyme HCCS was able to catalyze a covalent heme linkage to all of the substituted peptides except for SXXSH. By contrast, the bacterial synthase showed a more stringent thiol requirement, only reacting effectively with one of the homocysteine substitutions. Even here there were differences between the two enzymes, with the bacterial and mitochrondrial enzymes forming 2 and 1 thioether linkages, respectively.

In summary, the quality of the work is high, and it is clearly described. The in vitro reconstitution of holo-cytochrome c synthesis opens up numerous avenues to explore. The identified differences between the human and bacterial enzymes has mechanistic and evolutionary implications.

This study is novel, thorough and well-written and so I have no major suggestions.

---

## [Author Response]

Reviewer #1:This is a thoughtful and elegant study of how human (HCCS) and bacterial (CcsBA) cytochrome c synthases provide heme and catalyze covalent attachment of heme to its cognate cytochrome c recipient. The study is comprehensive and is well-done. I have only a few suggestions to help improve the readability of this manuscript and its overall impact.1. The authors have performed a comprehensive analyses using physicochemical and spectroscopic methods comparing their assay-generated holo cyt c to purified cyt c. Is there a functional assay to evaluate whether the holo cyt c generated in their assays are actually biologically active?

As reviewer #2 indicates, “functionality (is) evaluated by redox titrations”. The cyt c synthesized in vitro had identical potential as the published (and in vivo produced here).

2. I had a difficult time seeing the figures and reading some of the figure legends (Figure 2 is especially poorly-written).

We have divided up and enlarged figures to increase readability. We have revised figure legends to improve clarity. Figures are divided as follows: Figure 1 into 2 parts (a-e and f-k), Figure 2 into 2 parts (a-c and d-h), Figure 3 into 2 parts (b-c and a, d-i) to make the figures more accessible to the reader. We thank the reviewer for this suggestion.

3. In Figure 2h, the authors first added the peptide (11, 16, and 20mers) before performing a competition with apocyt c to determine how well heme transfer versus release occurs with HCCS. Have the authors performed this experiment in the presence of increasing concentrations of apocytc? What will be effect of the peptides if apocyt c was added simultaneously with the peptide for "hemylation"?

We did not perform the exact experiments that the reviewer is asking for. The conditions presented in this experiment are 10mM HCCS (30% heme occupancy) 10mM peptide and 20 mM apo cyt c. We have clarified the methods to include the assay concentrations (10mM HCCS, 10mM peptide, 20mM Apo cyt c) (see lines 657-660 final manuscript file). Although not included in the manuscript, we have varied the amount of peptide in the experiment during the initial assay development. Peptide was titrated using 1, 3, 10 mM concentrations. Lower peptide concentrations resulted in increased cyt c yield. We interpret this to mean that when heme present in the synthase, cyt c will be made, thus peptide inhibition is due the peptide consuming heme. This is addressed as a possibility in the Discussion.

Reviewer #2:[…] The concerns outlined below should be easily addressed by the authors and would likely require minor edits to the text. Thus this work warrant publication in eLife with the following revisions.1. A claim of the paper is that helix 1 is necessary, yet no experiments were done to determine if the helix forms in the peptides. Has the formation of the helix in the peptides, in which it is proposed to form, been investigated (such as by CD or crystallization)? If not, can the conclusion be modified to address this?

We have not experimentally determined the structure of the 11-, 16-, and 20mer peptides. The structures in Figure 3 are derived from the crystal structure of human cytochrome c (PDB: 3ZCF), as noted in the Figure legend 3 (see lines 715-717 final manuscript file). As noted, PEP-FOLD programs were used to predict alpha helical structures of peptides, these are consistent with the secondary structures in native cyt c. We have added language to the text to make it more clear what caveats there are to the alpha helical structure of the peptides (see discussion). Regardless of the alpha helical secondary structure, recognition by HCCS clearly also requires key sidechains within this sequence (e.g. 20mer peptide with K6A, K8D, K9D, K14D substitutions and in vivo work on F11). Although these sidechains are likely recognized in the context of alpha helix 1 structure, the helical structure of this sequence does not impact our results or conclusions thereof.

2. For the 16mer, it appears that only the biotin-16mer (Figure 2a and Table S2) was used for the studies whereas the Biotin-20mer and the 20mer are listed. Does the Biotin-20mer behave identically to the 20mer? Discussion and confirmation of this in the text is important to show that the biotin tag doesn't affect the peptide structure and thus the recognition.

The biotin tag does not impact recognition, as demonstrated with key peptides +/- biotin. We did not have all peptides synthesized +/- biotin.

3. For the CcsBA system the time necessary for in vitro cytochrome c biogenesis is 3 hours while the HCCs is 1 hour. Why the difference? Could the system be a missing factor? This could be addressed within the text.

The reviewer has made a point that may turn out interesting in the future. The synthesis over time for CcsBA (old Figure 3g, new Figure 6e) shows that after 1 hr, significant cyt c has heme attached. Although it takes another 2 hrs to reach maximum, the timeframe compared to HCCS is not indicative of a missing factor. More likely, it has to do with the different proteins themselves (CcsBA vs HCCS). For example, we could speculate that some of the heme attached in CcsBA may be coming from the internal TM-heme site that is transported into the P-heme site during the reaction. This will take significantly more investigation.

4. Statistical information is missing. While quantification on a number of experiments was performed, no statistical analysis is included. One example is in the results lines 285 to 286 and discussion lines 484 to 487, the release of cytochrome c, the 20mer and the 20mer SXXCH are discussed and data presented in Figures 2e and S4. While multiple experiments were performed no statistical analysis was provided. This should be added to understand the accuracy of these measurements and support for the conclusion.

The standard deviation has been added to new Figures 2c, 4c, and Figure 3—figure supplement 3. The text has also been changed to reflect these numbers and their standard deviations.

5. Based on the lack of release of peptides by HCCS it is concluded that the folding of the protein is necessary for optimal release. If helix 1 is formed and necessary for attachment, this suggests that some secondary structural elements are formed and that tertiary structure is necessary for release. As noted above investigation of the structural elements of the peptides would strengthen and this could be added to the text.

We considered a lengthy discussion on the features necessary for release—for example, it could be different secondary structures that “pull” the heme attached peptide out via specific interactions (eg the C-term alpha helix interacts with alpha helix 1 as a folding intermediate), or it could be the final folding into 3D, or even the Met81 ligand interaction in the folding pathway. We decided that these would be speculative, and this release step will require significantly more study for comment.

6. In the Results section there is no summary of the H and K mutants, it would be helpful to add one or two sentences as this is a significant part of the discussion and basis for the model presented in Figure 5.

We have moved the description of the H mutants to the Results section. We have chosen to leave the K mutants in the discussion as they specifically test the electrostatic hypothesis proposed in Figure 8. We felt that discussion of the K mutant prior to this would be a point of confusion for readers.